# Semantic Image Inversion and Editing using Rectified Stochastic Differential Equations

**Litu Rout**[1,2*]   **Yujia Chen**[1]   **Nataniel Ruiz**[1]
**Constantine Caramanis**[2]   **Sanjay Shakkottai**[2]   **Wen-Sheng Chu**[1]

[1] Google    [2] UT Austin

{litu.rout,constantine,sanjay.shakkottai}@utexas.edu
{liturout,yujiachen,natanielruiz,wschu}@google.com

## ABSTRACT

Generative models transform random noise into images, while their inversion aims to reconstruct structured noise for recovery and editing. This paper addresses two key tasks: (i) *inversion* and (ii) *editing* of real images using stochastic equivalents of rectified flow models (*e.g*, Flux). While Diffusion Models (DMs) dominate the field of generative modeling for images, their inversion suffers from faithfulness and editability challenges due to nonlinear drift and diffusion. Existing DM inversion methods require costly training of additional parameters or test-time optimization of latent variables. Rectified Flows (RFs) offer a promising alternative to DMs, yet their inversion remains underexplored. We propose RF inversion using dynamic optimal control derived via a linear quadratic regulator, and prove that the resulting vector field is equivalent to a rectified stochastic differential equation. We further extend our framework to design a stochastic sampler for Flux. Our method achieves state-of-the-art performance in zero-shot inversion and editing, surpassing prior works in stroke-to-image synthesis and semantic image editing, with large-scale human evaluations confirming user preference. See our project page https://rf-inversion.github.io/ for code and demo.

## 1 INTRODUCTION

Vision generative models typically transform noise into images. Inverting such models, given a reference image, involves finding the structured noise that can regenerate the original image. Efficient inversion must satisfy two crucial properties. First, the structured noise should produce an image that is *faithful* to the reference image. Second, the resulting image should be easily *editable* using new prompts, allowing fine modifications over the image.

Diffusion Models (DMs) have become the mainstream approach for generative modeling of images (Sohl-Dickstein et al., 2015; Song & Ermon, 2019; Ho et al., 2020), excelling at sampling from high-dimensional distributions (Ramesh et al., 2021; Saharia et al., 2022; Ramesh et al., 2022; Rombach et al., 2022; Podell et al., 2023; Pernias et al., 2024). The sampling process follows a Stochastic Differential Equation known as reverse SDE (Anderson, 1982; Efron, 2011; Song et al., 2021b). Notably, these models can invert a given image. Recent advances in DM inversion have shown a significant impact on conditional sampling, such as stroke-to-image synthesis (Meng et al., 2022), image editing (Hertz et al., 2022; Mokady et al., 2023; Couairon et al., 2023; Rout et al., 2023a;b; 2024; Delbracio & Milanfar, 2023) and stylization (Hertz et al., 2023; Rout et al., 2025).

Despite its widespread usage, DM inversion faces critical challenges in *faithfulness* and *editability*. First, the stochastic nature of the process requires fine discretization of the reverse SDE (Ho et al., 2020; Song et al., 2021b), which increases expensive Neural Function Evaluations (NFEs). Coarse discretization, on the other hand, leads to less faithful outputs (Meng et al., 2022), even with deterministic methods like DDIM (Song et al., 2021a;b). Second, nonlinearities in the reverse trajectory introduce unwanted drift, reducing the accuracy of reconstruction (Karras et al., 2024). While existing methods enhance faithfulness by optimizing latent variables (Rout et al., 2024) or prompt embeddings (Mokady et al., 2023; Miyake et al., 2023), they tend to be less efficient, harder to edit,

---

*This work was done during an internship at Google.

Figure 1: Our method consists of three key steps: (i) Inversion – We map a (corrupted) image to a structured noise using our controlled forward ODE (8); (ii) Conditioning – We condition the reverse process on the desired text (e.g., "A bedroom"); and (iii) Generation – We then transform the structured noise into the desired image using our controlled reverse ODE (15). (Left) Inversion with rectified flows results in an atypical noise sample (●), while optimal control produces a typical noise sample (●). Our method interpolates between these two processes to generate a typical structured noise (●). (Right) We follow a similar strategy when converting noise back into an image.

and rely on complex attention processors to align with a given prompt (Hertz et al., 2022; Rout et al., 2024). These added complexities make such methods less suitable for real-world deployment.

For inversion and editing, we introduce a zero-shot conditional sampling algorithm using Rectified Flows (RFs) (Liu et al., 2022; Albergo & Vanden-Eijnden, 2023; Lipman et al., 2022; Esser et al., 2024), a powerful alternative to DMs. Unlike DMs, where sampling is governed by a reverse SDE, RFs use an Ordinary Differential Equation known as reverse ODE, offering advantages in both efficient training and fast sampling. We construct a controlled forward ODE, initialized from a given image, to generate the initial conditions for the reverse ODE. The reverse ODE is then guided by an optimal controller, obtained through solving a Linear Quadratic Regulator (LQR) problem. We prove that the resulting vector fields have a stochastic interpretation with an appropriate drift and diffusion. We evaluate RF inversion on stroke-to-image generation and image editing tasks, with additional qualitative results on cartoonization, object insertion, image generation, and content-style composition. Our method significantly improves photorealism in stroke-to-image generation, surpassing a state-of-the-art (SoTA) method (Mokady et al., 2023) by 89%, while maintaining faithfulness to the input stroke. In addition, we show that RF inversion outperforms DM inversion (Meng et al., 2022) in faithfulness by 4.7% and in realism by 13.8% on LSUN-bedroom dataset (Wang et al., 2017). Figure 1 shows a graphical illustration of our method RF-Inversion.

Our theoretical and practical contributions can be summarized as:

- We present an efficient inversion method for RF models, including Flux, that requires no additional training, latent optimization, prompt tuning, or complex attention processors.
- We develop a new vector field for RF inversion, interpolating between two competing objectives: consistency with a possibly corrupted input image, and consistency with the "true" distribution of clean images (§3.3). We prove that this vector field is equivalent to a rectified SDE that interpolates between the stochastic equivalents of these competing objectives (§3.4).
- We demonstrate the faithfulness and editability of RF inversion across three benchmarks: (i) LSUN-Bedroom, (ii) LSUN-Church, and (iii) SFHQ, on two tasks: stroke-to-image synthesis and image editing. In addition, we provide extensive qualitative results and conduct large-scale human evaluations to assess user preference metrics (§5).

Finally, we note that RF-Inversion has the interpretation of a *counterfactual sampler*, specifically as an instantiation of noise abduction (Pearl et al., 2016); please see Appendix B for discussion.

## 2 RELATED WORKS

**DM Inversion.** Diffusion models have become the mainstream approach for generative modeling, making DM inversion an exciting area of research (Meng et al., 2022; Couairon et al., 2023; Song et al., 2021b; Hertz et al., 2023; Mokady et al., 2023; Rout et al., 2024). Among training-free methods, SDEdit (Meng et al., 2022) adds noise to an image and uses the noisy latent as structured noise. For semantic image editing based on a given prompt, it simulates the standard reverse SDE starting from this structured noise. SDEdit requires no additional parameter training, latent variable optimization, or complex attention mechanisms. However, it is less faithful to the original image because adding noise in one step is equivalent to linear interpolation between the image and noise, while the standard reverse SDE follows a nonlinear path (Liu et al., 2022; Karras et al., 2022).

An alternate method, DDIM inversion (Song et al., 2021a;b), recursively adds predicted noise at each forward step and returns the final state as the structured noise (illustrated by $Y_t$ process in Figure 1(a)). However, DDIM inversion often deviates significantly from the original image due to nonlinearities in the drift and diffusion coefficients, as well as inexact score estimates (Mokady et al., 2023). To reduce this deviation, recent approaches optimize prompt embeddings (Mokady et al., 2023) or latent variables (Rout et al., 2024), but they have high time complexity. Negative prompt inversion (Miyake et al., 2023) speeds up the inversion process but sacrifices faithfulness. Methods like CycleDiffusion (Wu & De la Torre, 2023) and Direction Inversion (Ju et al., 2023) use inverted latents as references during editing, but they are either computationally expensive or not applicable to rectified flow models like Flux (Black Forest Labs, 2024) or SD3 (Esser et al., 2024).

**DM Editing.** Efficient inversion is crucial for real image editing. Once a structured noise is obtained by inverting the image, a new prompt is fed into the T2I generative model. Inefficient inversion often fails to preserve the original content and therefore requires complex editing algorithms. These editing algorithms can be broadly classified into (i) attention control, such as prompt-to-prompt (Hertz et al., 2022), plug-and-play (PnP) (Tumanyan et al., 2023), (ii) optimization-based methods like DiffusionCLIP (Kim et al., 2022), DiffuseIT (Kwon & Ye, 2023), STSL (Rout et al., 2024), and (iii) latent masking to edit user-provided regions of an image (Nichol et al., 2022; Huberman-Spiegelglas et al., 2024) or automatically extracted from the generative model (Couairon et al., 2023). We focus on efficient inversion, avoiding the need for complex editing algorithms.

**Challenges in RF Inversion.** Previous inversion or editing approaches have been tailored towards diffusion models and do not directly apply to SoTA rectified flow models like Flux (Black Forest Labs, 2024). This limitation arises because the network architecture of Flux is MM-DiT (Peebles & Xie, 2023), which is fundamentally different from the traditional UNet used in DMs (Ho et al., 2020; Song et al., 2021a;b). In MM-DiT, text and image information are entangled within the architecture itself, whereas in UNet, text conditioning is handled via cross-attention layers. Additionally, Flux primarily uses T5 text encoder, which lacks an aligned latent space for images, unlike CLIP encoders. Therefore, extending these prior methods to modern T2I generative models requires a thorough investigation. We take the first step by inverting and editing a given image using Flux.

**RF Inversion and Editing.** DMs (Ho et al., 2020; Song et al., 2021a; Rombach et al., 2022) traditionally outperform RFs (Lipman et al., 2022; Liu et al., 2022; Albergo & Vanden-Eijnden, 2023) in high-resolution image generation. However, recent advances have shown that RF models like Flux can surpass SoTA DMs in text-to-image (T2I) generation tasks (Esser et al., 2024). Despite this, their inversion and editing capabilities remain underexplored. In this paper, we introduce an efficient RF inversion method that avoids the need for training additional parameters (Hu et al., 2021; Ruiz et al., 2023), optimizing latent variables (Rout et al., 2024), prompt tuning (Mokady et al., 2023), or using complex attention processors (Hertz et al., 2022). While our focus is on inversion and editing, we also show that our framework can be easily extended to generative modeling.

**Filtering, Control and SDEs.** There is a rich literature on the connections between nonlinear filtering, optimal control and SDEs (Fleming & Rishel, 1975; Øksendal, 2003; Tzen & Raginsky, 2019; Zhang & Chen, 2022). These connections are grounded in the Fokker-Planck equation (Øksendal, 2003), which RF methods (Lipman et al., 2022; Liu et al., 2022; Albergo & Vanden-Eijnden, 2023; Albergo et al., 2023) heavily exploit in sampling. Our study focuses on rectified flows for conditional sampling, and shows that the resulting drift field also has an optimal control interpretation.

## 3 METHOD

### 3.1 PRELIMINARIES

In generative modeling, the goal is to sample from a target distribution $p_0$ given a finite number of samples from that distribution. Rectified flows (Lipman et al., 2022; Liu et al., 2022) represent a class of generative models that construct a source distribution $q_0$ and a time varying vector field $v_t(\mathbf{x}_t)$ to sample $p_0$ using an ODE:

$$\mathrm{d}X_t = v_t(X_t)\mathrm{d}t, \quad X_0 \sim q_0, \quad t \in [0, 1]. \tag{1}$$

Starting from $X_0 = \mathbf{x}_0$, the ODE (1) is integrated from $t : 0 \to 1$ to yield a sample $\mathbf{x}_1$ distributed according to $p_0$ (e.g., the distribution over images). A common choice of $q_0$ is standard Gaussian $\mathcal{N}(0, I)$ and $v_t(X_t) = -u(X_t, 1 - t; \varphi)$, where $u$ is a neural network parameterized by $\varphi$. The neural network is trained using the conditional flow matching objective as discussed below.

**Training Rectified Flows.** To train a neural network to serve as the vector field for the ODE (1), we couple samples from $p_0$ with samples from $q_0$ – which we call $p_1$ to simplify the notation – via a linear path: $Y_t = tY_1 + (1-t)Y_0$. The resulting marginal distribution of $Y_t$ becomes:

$$p_t(\mathbf{y}_t) = \mathbb{E}_{Y_1 \sim p_1}[p_t(\mathbf{y}_t|Y_1)] = \int p_t(\mathbf{y}_t|\mathbf{y}_1)p_1(\mathbf{y}_1)\mathrm{d}\mathbf{y}_1. \tag{2}$$

Given an initial state $Y_0 = \mathbf{y}_0$ and a terminal state $Y_1 = \mathbf{y}_1$, the linear path induces an ODE: $\mathrm{d}Y_t = u_t(Y_t|\mathbf{y}_1)\mathrm{d}t$ with the conditional vector field $u_t(Y_t|\mathbf{y}_1) = \mathbf{y}_1 - \mathbf{y}_0$. The marginal vector field is derived from the conditional vector field using the following relation (Lipman et al., 2022):

$$u_t(\mathbf{y}_t) = \mathbb{E}_{Y_1 \sim p_1}\left[u_t(\mathbf{y}_t|Y_1)\frac{p_t(\mathbf{y}_t|Y_1)}{p_t(\mathbf{y}_t)}\right] = \int u_t(\mathbf{y}_t|\mathbf{y}_1)\frac{p_t(\mathbf{y}_t|\mathbf{y}_1)}{p_t(\mathbf{y}_t)}p_1(\mathbf{y}_1)\mathrm{d}\mathbf{y}_1. \tag{3}$$

We can then use a neural network $u(\mathbf{y}_t, t; \varphi)$, parameterized by $\varphi$, to approximate the marginal vector field $u_t(\mathbf{y}_t)$ through the flow matching objective defined as:

$$\mathcal{L}_{FM}(\varphi) \coloneqq \mathbb{E}_{t \sim \mathcal{U}[0,1], Y_t \sim p_t}\left[\|u_t(Y_t) - u(Y_t, t; \varphi)\|_2^2\right]. \tag{4}$$

For tractability, we can instead consider a different objective, called conditional flow matching:

$$\mathcal{L}_{CFM}(\varphi) \coloneqq \mathbb{E}_{t \sim \mathcal{U}[0,1], Y_t \sim p_t(\cdot|Y_1), Y_1 \sim p_1}\left[\|u_t(Y_t|Y_1) - u(Y_t, t; \varphi)\|_2^2\right]. \tag{5}$$

$\mathcal{L}_{CFM}$ and $\mathcal{L}_{FM}$ have the identical gradients (Lipman et al., 2022, Theorem 2), and are hence equivalent. However, $\mathcal{L}_{CFM}(\varphi)$ is computationally tractable, unlike $\mathcal{L}_{FM}(\varphi)$, and therefore preferred during training. Finally, the required vector field in (1) is computed as $v_t(X_t) = -u(X_t, 1-t; \varphi)$. In this way, rectified flows sample a data distribution by an ODE with a learned vector field.

## 3.2 Connection between Rectified Flows and Linear Quadratic Regulator

The unconditional rectified flows (RFs) (e.g., Flux) from Section §3.1 above, enable image generation by simulating the vector field $v_t(\cdot)$ initialized with a sample of random noise. Subsequently, by simulating the reversed vector field $-v_{1-t}(\cdot)$ starting from the image, we get back the sample of noise that we started with. We formalize this statement below.

**Proposition 3.1.** *Given an image $\mathbf{y}_0$ and the vector field $v_t(\cdot)$ of the generative ODE (1), suppose the structured noise $\mathbf{y}_1$ is obtained by simulating an ODE:*

$$\mathrm{d}Y_t = u_t(Y_t)\mathrm{d}t, \quad Y_0 = \mathbf{y}_0, \quad t \in [0,1]. \tag{6}$$

*If $u_t(\cdot) = -v_{1-t}(\cdot)$ and $X_0 = \mathbf{y}_1$, then the ODE (1) recovers the original image, i.e., $X_1 = \mathbf{y}_0$.*

**Implication.** Rectified flows enable exact inversion of a given image when the vector field of the generative ODE (1) is precisely known. Employing ODE (6) for the structured noise and ODE (1) to transform that noise back into an image, RF inversion accurately recovers the given image.

Suppose instead that we start with a *corrupted image* and simulate the reversed vector field $-v_{1-t}(\cdot)$. Then we obtain a noise sample. There are two salient aspects of this noise sample. First, it is consistent with the original image: when processed through $v_t(\cdot)$ it results in the same corrupted image. Second, if the image sample is "atypical" (e.g., corrupted, or, say, a stroke painting as in §5), then the sample of noise is also likely to be atypical. In other words, the noise sample is only consistent to the (possibly corrupted) image sample.

Our goal is to modify the pipeline above so that even when we start with a corrupted image, we can get back a clean image (see stroke-to-image synthesis in Figure 3), but for this, we need to process by $v_t(\cdot)$ a noise sample that is closer to being "typical".

Thus, as a first step, we derive an optimal controller that takes a minimum energy path to convert any image $Y_0$ (whether corrupted or not) to a typical noise sample $Y_1 \sim p_1$ (Fleming & Rishel, 1975). Specifically, we consider optimal control in a $d$-dimensional vector space $\mathbb{R}^d$:

$$V(c) \coloneqq \int_0^1 \frac{1}{2}\|c(Z_t, t)\|_2^2 \, \mathrm{d}t + \frac{\lambda}{2}\|Z_1 - Y_1\|_2^2, \; \mathrm{d}Z_t = c(Z_t, t)\,\mathrm{d}t, \; Z_0 = \mathbf{y}_0, \; Y_1 \sim p_1, \tag{7}$$

where $\lambda$ is the weight assigned to the terminal cost and $V(c)$ denotes the total cost of the control $c : \mathbb{R}^d \times [0,1] \to \mathbb{R}^d$. The minimization of $V(c)$ over the admissible set of controls, denoted by $\mathcal{C}$, is known as the Linear Quadratic Regulator (LQR) problem. The solution of the LQR problem (7) is given in **Proposition 3.2**, which minimizes the quadratic transport cost of the dynamical system.

**Proposition 3.2.** *For $Z_0 = \mathbf{y}_0$ and $Y_1 = \mathbf{y}_1$, the optimal controller of the LQR problem (7), denoted by $c^*(\cdot, t)$ is equal to the conditional vector field $u_t(\cdot|\mathbf{y}_1)$ of the rectified linear path $Y_t = tY_1 + (1-t)Y_0$ when $Y_0 = \mathbf{y}_0$, i.e., $c^*(\mathbf{z}_t, t) = u_t(\mathbf{z}_t|\mathbf{y}_1) = (\mathbf{y}_1 - \mathbf{z}_t)/(1-t)$.*

### 3.3 INVERTING RECTIFIED FLOWS WITH DYNAMIC CONTROL

So far, we have two vector fields. The first, from the RFs, transforms an image $Y_0$ typical for distribution $p_0$ to a typical sample of random Gaussian noise $Y_1 \sim p_1$. As discussed above, if the image sample is atypical, then the sample of noise is also likely to be atypical. We also have a second vector field resulting from the optimal control formulation that transforms *any* image (whether corrupted or not) to a noise sample that is typical-by-design from the distribution $p_1$. Therefore, this sample, when passed through the rectified flow ODE (1) results in a "typical" image from the "true" distribution $p_0$. This image is clean, i.e., typical for $p_0$, but it is not related to the image $Y_0$. Our controlled ODE, defined below, interpolates between these two differing objectives – *consistency with the given (possibly corrupted) image, and consistency with the distribution of images $p_0$* – with a tunable parameter $\gamma$:

$$\mathrm{d}Y_t = \Big[u_t(Y_t) + \gamma\,(u_t(Y_t|\mathbf{y}_1) - u_t(Y_t))\Big]\mathrm{d}t, \quad Y_0 = \mathbf{y}_0, \tag{8}$$

where $u_t(Y_t|\mathbf{y}_1) = c^*(Y_t, t)$ is computed based on the insights from **Proposition 3.2**, and $u_t(Y_t) = -v_{1-t}(Y_t)$ as established in **Proposition 3.1**. Here, we call $\gamma \in [0, 1]$ the *controller guidance*. Thus, ODE (8) generalizes (6) to editing applications, while keeping its inversion accuracy comparable.

When $\gamma = 1$, the drift field of the ODE (8) becomes optimal controller of LQR problem (7), ensuring that the structured noise $Y_1 = \mathbf{y}_1$ adheres to the distribution $p_1$. Consequently, initializing the generative ODE (1) with $\mathbf{y}_1$ results in samples with high likelihood under the data distribution $p_0$.

Conversely, when $\gamma = 0$, the system follows the ODE (6) described in **Proposition 3.1**, resulting a structured noise $Y_1$ that is not guaranteed to follow the noise distribution $p_1$. However, initializing the generative ODE (1) with this noise precisely recovers the reference image $\mathbf{y}_0$.

Beyond this vector field interpolation intuition, we show in the next section §3.4 that the controlled ODE (8) has an SDE interpretation. As is well known (Ho et al., 2020; Song et al., 2021a; Meng et al., 2022; Song et al., 2021b), SDEs are robust to initial conditions, in proportion to the variance of the additive noise. Specifically, errors propagate over time in an ODE initialized with an incorrect or corrupted sample. However, SDEs (Markov processes) under appropriate conditions converge to samples from a carefully constructed invariant distribution with reduced sensitivity to the initial condition, resulting in a form of robustness to initialization. As we see, the parameter $\gamma$ (the controller guidance) appears in the noise term to the SDE, thus the SDE analysis in the next section again provides intuition on the trade-off between consistency to the (corrupted) image and consistency to the terminal invariant distribution, and helps design a stochastic sampler for Flux (Appendix D.7).

**Remark 3.3.** *We note that our analysis extends to the case where $\gamma$ is time-varying, though we omit these results for simplicity of notation. This is useful in practice, especially when $\mathbf{y}_0$ is a corrupted image, because for large $\gamma$ the stochastic evolution (22) moves toward a sample from the invariant measure $\mathcal{N}(0, I)$. This noise encodes clean images. Starting from this noise, the corresponding reverse process operates in pure diffusion mode, resulting in a clean image. As the process approaches the terminal state, $\gamma$ is gradually reduced to ensure that $\mathbf{y}_0$ is encoded through $u_t(\cdot)$ into the final structured noise sample.*

### 3.4 CONTROLLED RECTIFIED FLOWS AS STOCHASTIC DIFFERENTIAL EQUATIONS

An SDE (Ho et al., 2020) is known to have an equivalent ODE formulation (Song et al., 2021a) under certain regularity conditions (Anderson, 1982; Song et al., 2021b). In this section, we derive the opposite: an SDE formulation for our controlled ODE (8) from §3.3. Let $W_t$ be a $d$-dimensional Brownian motion in a filtered probability space $(\Omega, \mathcal{F}, \{\mathcal{F}_t\}, \mathbb{P})$.

**Theorem 3.4.** *Fix any $T \in (0, 1)$. For any $t \in [0, T]$, the controlled ODE (8) is explicitly given by:*

$$\mathrm{d}Y_t = \left[-\frac{1}{1-t}\,(Y_t - \gamma\mathbf{y}_1) - \frac{(1-\gamma)t}{1-t}\nabla \log p_t(Y_t)\right]\mathrm{d}t, \quad Y_0 \sim p_0. \tag{9}$$

*Its density evolution is identical to the density evolution of the following SDE for $Y_0 \sim p_0$:*

$$\mathrm{d}Y_t = \left(-\frac{1}{1-t}(Y_t - \gamma \mathbf{y}_1) - \frac{t}{1-t}(1-\gamma)\left(\nabla \log p_t(Y_t) - \nabla \log p_{t,\gamma}(Y_t)\right)\right)\mathrm{d}t + \sqrt{\frac{2(1-\gamma)t}{1-t}}\mathrm{d}W_t.$$
(10)

*Finally, denoting $p_{t,\gamma}(\cdot)$ as the marginal pdf of $Y_t$, the density evolution is explicitly given by:*

$$\frac{\partial p_{t,\gamma}(Y_t)}{\partial t} = \nabla \cdot \left[\left(\frac{1}{1-t}(Y_t - \gamma \mathbf{y}_1) + \frac{(1-\gamma)t}{1-t}\nabla \log p_t(Y_t)\right)p_{t,\gamma}(Y_t)\right].$$
(11)

**Properties of SDE (10).** Elaborating on the intuition discussed at the end of §3.3, when the controller guidance $\gamma = 0$, then $\nabla \log p_t(Y_t) = \nabla \log p_{t,0}(Y_t)$, and SDE (10) becomes the stochastic equivalent of the standard RFs; see **Lemma A.2**. The resulting SDE is given by

$$\mathrm{d}Y_t = -\frac{1}{1-t}Y_t \mathrm{d}t + \sqrt{\frac{2t}{1-t}}\mathrm{d}W_t, \quad Y_0 \sim p_0,$$
(12)

which improves faithfulness to the image $Y_0$. When $\gamma = 1$, the SDE (10) solves the LQR problem (7) and drives towards the terminal state $Y_1 = \mathbf{y}_1$. This improves the generation quality, because the sample $Y_1$ is from the correct noise distribution $p_1$ as previously discussed in §3.3. Therefore, a suitable choice of $\gamma$ retains faithfulness while simultaneously applying the desired edits.

Finally, we assume $T = 1 - \delta$ for sufficiently small $\delta$ (such that $0 < \delta \ll 1$) to avoid irregularities at the boundary (Øksendal, 2003). This is typically considered in practice for numerical stability (even for diffusion models). Thus, in practice, the final sample $\mathbf{y}_{1-\delta}$ is returned as $\mathbf{y}_1$.

**Comparison with DMs.** Analogous to the SDE (12), the stochastic noising process of DMs is typically modeled by the Ornstein-Uhlenbeck (OU) process, governed by the following SDE:

$$\mathrm{d}Y_t = -Y_t \mathrm{d}t + \sqrt{2}\mathrm{d}W_t.$$
(13)

The corresponding ODE formulation is given by:

$$\mathrm{d}Y_t = [-Y_t - \nabla \log p_t(Y_t)]\mathrm{d}t.$$
(14)

Instead, our approach is based on rectified flows (1), which leads to a different ODE and consequently translates into a different SDE. As an additional result, we formalize the ODE derivation in **Lemma A.1**. In **Lemma A.2**, we show that the marginal distribution of this ODE is equal to that of an SDE with appropriate drift and diffusion terms. In **Proposition A.3**, we show that the stationary distribution of this new SDE (12) converges to the standard Gaussian $\mathcal{N}(0, I)$ in the limit as $t \to 1$.

The standard OU process (13) interpolates between the data distribution at time $t = 0$ and a standard Gaussian as $t \to \infty$. The SDE (12), however, interpolates between the data distribution at time $t = 0$ and a standard Gaussian at $t = 1$. In other words, it effectively "accelerates" time as it progresses to achieve the terminal Gaussian distribution. This is accomplished by modifying the coefficients of drift and diffusion as in (12) to depend explicitly on time $t$. Thus, a sample path of (12) appears like a noisy line, unlike that of the OU process (see Appendix D.3 for numerical simulations).

## 3.5 Controlled Reverse Flow using Rectified ODEs and SDEs

In this section, we develop an ODE and an SDE similar to our discussions above, but for the reverse direction (i.e., from noise to images).

**Reverse process using ODE.** Starting from the structured noise $\mathbf{y}_1$ obtained by integrating the controlled ODE (8), we construct another controlled ODE (15) for the reverse process (i.e., noise to image). In this process, the optimal controller uses the reference image $\mathbf{y}_0$ for guidance:

$$\mathrm{d}X_t = \left[v_t(X_t) + \eta\left(v_t(X_t | \mathbf{y}_0) - v_t(X_t)\right)\right]\mathrm{d}t, \quad X_0 = \mathbf{y}_1, \quad t \in [0,1],$$
(15)

where $\eta \in [0,1]$ is the *controller guidance parameter* as before that controls faithfulness and editability of the given image $\mathbf{y}_0$. Similar to the analysis in **Proposition 3.2**, $v_t(X_t | \mathbf{y}_0)$ is obtained by solving the modified LQR problem (16):

$$V(c) = \int_0^1 \frac{1}{2}\|c(Z_t, t)\|_2^2 \mathrm{d}t + \frac{\lambda}{2}\|Z_1 - \mathbf{y}_0\|_2^2, \quad \mathrm{d}Z_t = c(Z_t, t)\mathrm{d}t, \quad Z_0 = \mathbf{y}_1.$$
(16)

Solving (16), we get $c(Z_t, t) = \frac{\mathbf{y}_0 - Z_t}{1-t}$. Our controller steers the samples toward the given image $\mathbf{y}_0$. Thus, the controlled reverse ODE (15) effectively reduces the reconstruction error incurred in the standard reverse ODE (1) of RF models (e.g. Flux).

**Reverse process using SDE.** Finally, in **Theorem 3.5**, we provide the stochastic equivalent of our controlled reverse ODE (15) for generation. Recall that we initialize with the terminal structured noise by running the controlled forward ODE (8), along with a reference image $\mathbf{y}_0$. As discussed above, we terminate the inversion process at a time $T = 1 - \delta$ for numerical stability, resulting in a vector $\mathbf{y}_{1-\delta}$. Our reverse SDE thus starts at a corresponding time $\delta$ with this vector $\mathbf{y}_{1-\delta}$ at initialization, and terminates at time $T' < 1$.

**Theorem 3.5.** *Fix any $T' \in (\delta, 1)$, and for any $t \in [\delta, T']$, the density evolution of the controlled ODE (15) initialized at $X_0 = \mathbf{y}_{1-\delta}$ is identical to the density evolution of the following SDE:*

$$\mathrm{d}X_t = \left[ \frac{(1-t-\eta)X_t + \eta t \mathbf{y}_0}{t(1-t)} + \frac{(1-t)(1-\eta)}{t} \Big( \nabla \log p_{1-t}(X_t) + \nabla \log q_{t,\eta}(X_t) \Big) \right] \mathrm{d}t \quad (17)$$

$$+ \sqrt{\frac{2(1-t)(1-\eta)}{t}} \mathrm{d}W_t.$$

*Furthermore, denoting $q_t(\cdot)$ as the marginal pdf of $X_t$, its density evolution is given by:*

$$\frac{\partial q_{t,\eta}(X_t)}{\partial t} = \nabla \cdot \left[ - \Big( \frac{1-t-\eta}{t(1-t)} X_t + \frac{\eta}{1-t} \mathbf{y}_0 + \frac{(1-t)}{t}(1-\eta)\nabla \log p_{1-t}(X_t) \Big) q_{t,\eta}(Y_t) \right]. \quad (18)$$

**Properties of SDE (17).** When the controller parameter $\eta = 0$, we obtain a stochastic sampler (22) for the pre-trained Flux, as given in **Lemma A.4**. This case of our SDE (17) corresponds to the stochastic variant of standard RFs (Liu et al., 2022; Lipman et al., 2022; Albergo & Vanden-Eijnden, 2023). Our key contribution lies in conditioning on $X_1 = \mathbf{y}_0$ for inverting rectified flows. Importantly, our explicit construction does not require additional training or test-time optimization, enabling for the first time an efficient sampler for zero-shot inversion and editing using Flux. When $\eta = 1$, the score term and Brownian motion vanish from the SDE (17). The resulting drift becomes $\frac{\mathbf{y}_0 - X_t}{1-t}$, the optimal controller for the LQR problem (16), exactly recovering the given image $\mathbf{y}_0$.

**Remark 3.6.** *Similar to **Remark 3.3**, our analysis extends to the case when $\eta$ is time-varying. This is useful in editing, as it allows the flow to initially move toward the given image $\mathbf{y}_0$ by choosing a large $\eta$. As the flow approaches $\mathbf{y}_0$ on the image manifold, $\eta$ is gradually reduced, ensuring that the text-guided edits are enforced through the unconditional vector field $v_t(\cdot)$ provided by Flux.*

Finally, we note that RF-Inversion has an interpretation as a counterfactual sampler; please see Appendix B for more discussion on this interpretation (Pearl et al., 2016).

## 4 ALGORITHM: INVERSION AND EDITING VIA CONTROLLED ODEs

In this section, we define the problem setup and outline the procedure using controlled ODEs (8) and (15). We employ **Algorithm 1** for inversion and **Algorithm 2** for editing: see Appendix D.

**Problem Setup.** The user provides a text "prompt" to edit reference content, which could be a corrupt or a clean image. For the corrupt image guide, we use the dataset from SDEdit (Meng et al., 2022), which contains color strokes to convey high-level details. In this setting, the reference guide $\mathbf{y}_0$ is typically not a realistic image under the data distribution $p_0$. The objective is to transform this guide into a more realistic image under $p_0$ while maintaining faithfulness to the original guide.

For the clean image guide, the user provides a real image $\mathbf{y}_0$ along with an accompanying text "prompt" to specify the desired edits. The task is to apply text-guided edits to $\mathbf{y}_0$ while preserving its content. Examples include face editing, where the text might instruct change in age or gender.

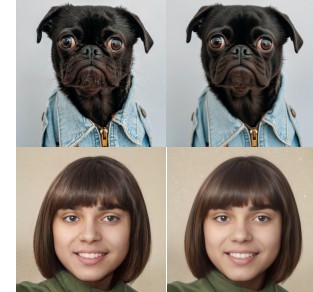

Original      RF inversion

Figure 2: Inverting flows by controlled ODEs (8) and (15).

**Procedure.** Our algorithm has two key steps: **inversion** and **editing**. We discuss each step below.

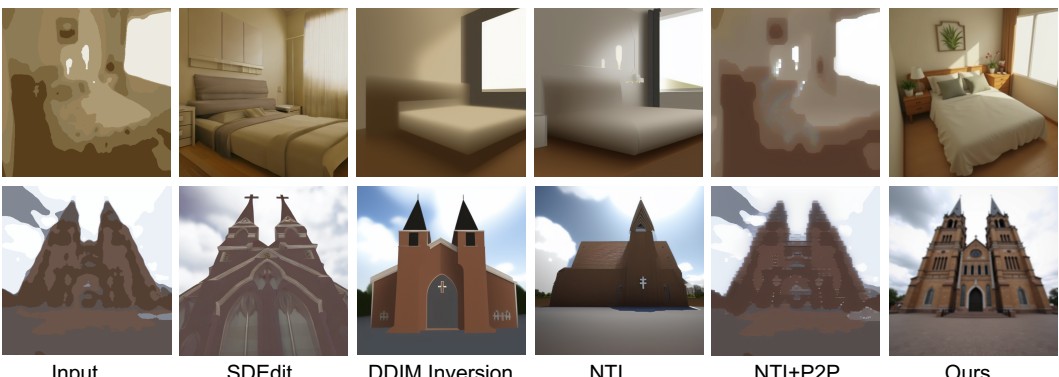

| Input | SDEdit | DDIM Inversion | NTI | NTI+P2P | Ours |

Figure 3: **Stroke2Image generation.** Our method generates photo-realistic images of bedroom or church given stroke paints, showing robustness to initial corruptions.

**Inversion.** The first step involves computing the structured noise $Y_1$ by employing our controlled ODE (8), initialized at the reference content $Y_0 = \mathbf{y}_0$. To compute the unconditional vector field, we use the pre-trained Flux model $u(\cdot, \cdot, \cdot; \varphi)$, which requires three inputs: the state $Y_t$, the time $t$, and the prompt embedding $\Phi(\text{prompt})$. During the inversion process, we use null prompt in the Flux model, i.e., $u_t(\mathbf{y}_t) = u(\mathbf{y}_t, t, \Phi(\text{""}); \varphi)$. For the conditional vector field, we apply the analytical solution derived in **Proposition 3.2**. The inversion process yields a latent variable that is then used to initialize our controlled ODE (15), i.e., $X_0 = \mathbf{y}_1$. In this phase, we again use the null prompt to compute the vector field $v_t(\mathbf{x}_t) = -u(\mathbf{x}_t, 1-t, \Phi(\text{""}); \varphi)$: see Figure 2 for the final output.

**Editing.** The second step involves text-guided editing of the reference content $\mathbf{y}_0$. This process is governed by our controlled ODE (15), where the vector field is computed using the desired text prompt within Flux: $v_t(X_t) = -u(\mathbf{x}_t, 1-t, \Phi(\text{prompt}); \varphi)$. The controller guidance $\eta$ in (15) balances faithfulness and editability: higher $\eta$ improves faithfulness but limits editability, while lower $\eta$ allows significant edits at the cost of reduced faithfulness. Consequently, the controller guidance $\eta$ provides a smooth interpolation between faithfulness and editability, a crucial feature in semantic image editing. Motivated by **Remark 3.3** and **3.6**, we consider a time-varying controller guidance $\eta_t$, such that for a fixed $\eta \in [0, 1]$ and $\tau \in [0, 1]$, $\eta_t = \eta \ \forall t \leq \tau$ and 0 otherwise.

## 5 EXPERIMENTAL EVALUATION

We show that RF inversion outperforms DM inversion across three benchmarks: LSUN-church, LSUN-bedroom (Wang et al., 2017), and SFHQ (Beniaguev, 2022) on two tasks: Stroke2Image generation and image editing. Stroke2Image generation shows the robustness of our algorithm to initial corruption. In semantic image editing, we emphasize the ability to edit clean images without additional training, optimization, or complex attention processors. We use Flux (Black Forest Labs, 2024) as our base foundation model and show compatibility with SD3.5 (Esser et al., 2024).

**Baselines.** As this paper focuses on inverting flows, we compare with SoTA inversion approaches, such as NTI (Mokady et al., 2023), DDIM Inversion (Song et al., 2021a), and SDEdit (Meng et al., 2022). We use the official NTI implementation for both NTI and DDIM inversion, and Diffusers library for SDEdit. Hyper-parameters for all these baselines are tuned for optimal performance. We compare with NTI for both direct prompt change and with prompt-to-prompt Hertz et al. (2022) editing. All methods are training-free; however, NTI (Mokady et al., 2023) solves an optimization problem at each denoising step during inversion and uses P2P (Hertz et al., 2022) attention processor during editing. We follow the evaluation protocol from SDEdit (Meng et al., 2022). More qualitative results and comparison are in Appendix §D.

**Stroke2Image generation.** As discussed in §4, our goal is to generate a photo-realistic image from a stroke paint (a corrupted image) and the text prompt "photo-realistic picture of a bedroom". In this case, the high level details in the stroke painting guide the reverse process toward a clean image.

In Figure 3, we compare RF inversion (ours) with DM inversions. DM inversions propagate the corruption from the stroke painting into the structured noise, which leads to outputs resembling the input stroke painting. NTI optimizes null embeddings to align the reverse process with the DDIM forward trajectory. Although adding P2P to the NTI pipeline helps localized editing as in Figure 4,

Table 1: **Quantitative results for Stroke2Image generation.** L2 and Kernel Inception Distance (KID) capture faithfulness and realism, respectively. Optimization-based methods are colored gray. User Pref. shows the percentage of users that prefer our method over each alternative in pairwise comparisons (and ties). E.g.: 62.11% (+ 8% ties) prefer ours over SDEdit-Flux for LSUN Bedroom.

| Method | LSUN Bedroom | | | LSUN Church | | |
| --- | --- | --- | --- | --- | --- | --- |
| | L2 ↓ | KID ↓ | User Pref. (%) ↑ | L2 ↓ | KID ↓ | User Pref. (%) ↑ |
| SDEdit-SD1.5 | 86.72 | 0.029 | 59.67 (5.33) | 90.72 | 0.089 | 65.33 (4.11) |
| SDEdit-SDXL | 96.82 | 0.133 | - | 98.19 | 0.112 | - |
| SDEdit-SD3 | 93.40 | 0.037 | - | 98.14 | 0.096 | - |
| SDEdit-Flux | 94.89 | 0.032 | 62.11 (8.00) | 92.47 | 0.081 | 66.22 (5.22) |
| DDIM Inv-SD1.5 | 87.95 | 0.113 | 82.56 (1.67) | 97.36 | 0.107 | 85.44 (2.78) |
| NTI-SD1.5 | 82.77 | 0.095 | 80.89 (4.33) | 87.88 | 0.098 | 77.11 (4.89) |
| NTI+P2P-SD1.5 | 46.46 | 0.234 | 98.11 (1.78) | 34.48 | 0.168 | 99.22 (0.56) |
| Ours | 82.65 | 0.025 | - | 80.36 | 0.059 | - |

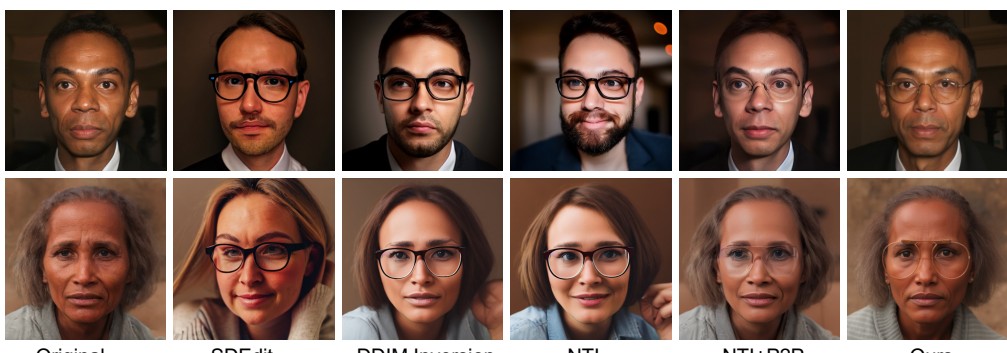

Original     SDEdit     DDIM Inversion     NTI     NTI+P2P     Ours

Figure 4: **Image editing for adding face accessories.** Prompt: "face of a man/woman wearing glasses". The proposed method better preserves the identity while applying the desired edits.

for corrupted images, it drives the reverse process even closer to the corruption. In contrast, our controlled ODE (8) yields a structured noise that is consistent with the corrupted image and also the invariant terminal distribution, as discussed in §3.3, resulting in more realistic images.

In Table 1, we show that our method outperforms prior works in faithfulness and realism. On the test split of LSUN bedroom dataset, our approach is 4.7% more faithful and 13.79% more realistic than the best optimization free method SDEdit-SD1.5. Ours is 73% more realistic than the optimization-based method NTI, but comparable in L2. As discussed, NTI+P2P gets closer toward the corrupt image, which gives a very low L2 error, but the resulting image becomes unrealistic. Our approach is 89% more realistic than NTI+P2P. We observe similar gains on LSUN church dataset.

**User study.** We conduct a user study using Amazon Mechanical Turk to evaluate the overall performance of the our method. With 3 responses for each question, we collected in total 9,000 comparisons from 126 participants. As given in Table 1, our method outperforms all the other baselines by at least 59.67% in terms of overall satisfaction. More details are provided in Appendix §D.6.

**Semantic Image Editing.** Given a *clean image* and a text "prompt", the objective is to modify the image according to the given text while preserving the contents of the image (identity for face images). In rectified linear paths, editing from a noisy latent becomes straightforward, further enhancing the efficiency of our approach. Compared with SoTA approaches (Figure 4), our method requires no additional optimization or complex attention processors as in NTI (Mokady et al., 2023)+P2P(Hertz et al., 2022). Thus, it is more efficient than a current SoTA approach, and importantly, more faithful to the original image while applying the desired edits.

In Table 2, we show that our method outperforms the optimization-free methods by at least 29% in face reconstruction, 6.6% in DINO patch-wise similarity, and 26.4% in CLIP-Image similarity while being comparable in prompt alignment metric CLIP-T. Importantly, our approach offers 54.11% gain in runtime while staying comparable to NTI+P2P.

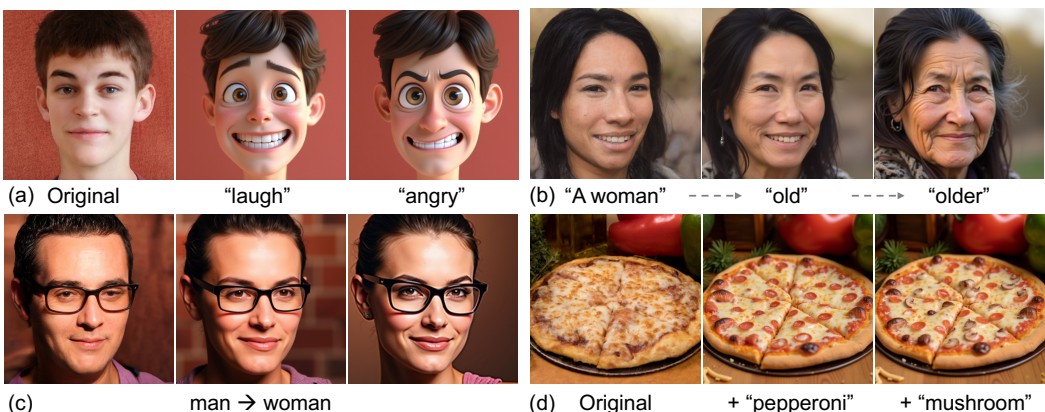

Figure 5: **Semantic Image Editing:** (a) stylized expression, (b) age, (c) gender, (d) object insert. Given an image and a text prompt, our algorithm performs semantic image editing in the wild.

Table 2: **Quantitative results for face editing** on SFHQ for "wearing glasses".

| Method | Face Rec. ↓ | DINO ↑ | CLIP-T ↑ | CLIP-I ↑ | Runtime(s) ↓ |
|---|---|---|---|---|---|
| SDEdit-SD1.5 | 0.626 | 0.885 | 0.300 | 0.712 | 8 |
| SDEdit-Flux | 0.632 | 0.892 | 0.292 | 0.710 | 24 |
| DDIM Inv. | 0.709 | 0.884 | 0.311 | 0.669 | 15 |
| NTI | 0.707 | 0.876 | 0.304 | 0.666 | 78 |
| NTI+P2P | 0.443 | 0.953 | 0.293 | 0.845 | 85 |
| Ours | 0.442 | 0.951 | 0.300 | 0.900 | 39 |

In Figure 5, we showcase four complex editing tasks: (a) prompt-based stylization with the prompt: "face of a boy in disney 3d cartoon style", where facial expressions, such as "laugh" or "angry" are used for editing; (b) ability to control the age of a person; (c) interpolating between two concepts: "A man" ↔ "A woman"; (d) sequentially inserting pepperoni and mushroom to an image of a pizza. We provide more examples of editing in the wild in Appendix §D.

**Comparison using the same backbone: Flux.** In Figure 6, we compare our method with SDEdit and DDIM inversion both adapted to Flux. NTI optimizes null embeddings to align with forward latents before applying text-guided edits, an approach well-suited for DMs that use both null and text embeddings. However, this strategy cannot be applied to Flux, as it does not explicitly use null embeddings. Consequently, we only reimplement SDEdit and DDIM inversion with the Flux backbone and compare them to our method. Since all methods leverage the same generative model, the improvements clearly stem from our controlled ODEs, grounded in a solid theoretical foundation (§3).

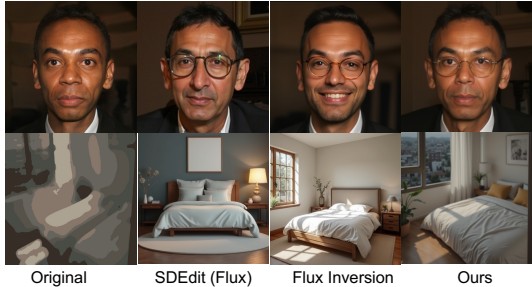

Figure 6: Comparison using Flux backbone.

## 6 CONCLUSION

We present the first approach for inversion and editing with state-of-art rectified flow models such as Flux. Our method interpolates between two vector fields: *(i)* the unconditional RF field that transforms a "clean" image to "typical" noise, and *(ii)* a conditional vector field derived from optimal control that transforms *any* image (clean or not) to "typical" noise. Our new field thus navigates between these two competing objectives of *consistency with the given (possibly corrupted) image, and consistency with the distribution of clean images*. Theoretically, we show that this is equivalent to a new rectified SDE formulation, sharing this intuition of interpolation. Practically, we show that our method results in state-of-art zero-shot performance, without the need of additional training, optimization of latent variables, prompt tuning, or complex attention processors.

## BROADER IMPACT STATEMENT

Semantic image inversion and editing have both positive and negative social impacts.

On the positive side, this technology enables (i) the generation of photo-realistic images from high level descriptions, such as stroke paintings, and (ii) the modification of clean images by changing various attributes like the age, gender, or adding glasses (§5).

On the negative side, it can be misused by malicious users to manipulate photographs of individuals with inappropriate or offensive edits. Additionally, it carries the inherent risks associated with the underlying generative model.

To mitigate the negative social impacts, we enable safety features such as NSFW filters in the underlying generative model. Furthermore, we believe watermarking images generated by this technology can reduce misuse in inversion and editing applications.

**Reproducibility.** The pseudocode and hyper-parameter details have been provided in the paper. Refer to our project page: https://rf-inversion.github.io/ for source code and demo.

## ACKNOWLEDGMENTS

This research has been supported by NSF Grant 2019844, a Google research collaboration award, and the UT Austin Machine Learning Lab. We thank Dr. Karthikeyan Shanmugam (Google DeepMind) for insightful discussions on the connections to causal inference and counterfactuals. We also thank the reviewers and area chair for their suggestions during the discussion phase.

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

## A  ADDITIONAL THEORETICAL RESULTS

In this section, we present the theoretical results omitted from the main draft due to space constraints. We formalize the ODE derivation of the standard rectified flows in **Lemma A.1**.

**Lemma A.1.** *Given a coupling $(Y_0, Y_1) \sim p_0 \times p_1$, consider the noising process $Y_t = tY_1 + (1 - t)Y_0$. Then, the rectified flow ODE formulation with the optimal vector field is given by*

$$\mathrm{d}Y_t = \left[ -\frac{1}{1-t}Y_t - \frac{t}{1-t}\nabla \log p_t(Y_t) \right] \mathrm{d}t, \quad Y_0 \sim p_0. \tag{19}$$

*Furthermore, denoting $p_t(\cdot)$ as the marginal pdf of $Y_t$, its density evolution is given by:*

$$\frac{\partial p_t(Y_t)}{\partial t} = \nabla \cdot \left[ \left( \frac{1}{1-t}Y_t + \frac{t}{1-t}\nabla \log p_t(Y_t) \right) p_t(Y_t) \right]. \tag{20}$$

In **Lemma A.2**, we show that the marginal distribution of the rectified flow (6) is equal to that of an SDE with appropriate drift and diffusion terms.

**Lemma A.2.** *Fix any $T \in (0, 1)$, and for any $t \in [0, T]$, the density evolution (20) of the rectified flow model (19) is identical to the density evolution of the following SDE:*

$$\mathrm{d}Y_t = -\frac{1}{1-t}Y_t \mathrm{d}t + \sqrt{\frac{2t}{1-t}}\mathrm{d}W_t, \quad Y_0 \sim p_0. \tag{21}$$

In **Proposition A.3**, we show that the stationary distribution of the SDE (21) converges to the standard Gaussian $\mathcal{N}(0, I)$ in the limit as $t \to 1$.

**Proposition A.3.** *Fix any $T \in (0, 1)$, and for any $t \in [0, T]$, the density evolution for the rectified flow ODE (6) is same as that of the SDE (12). Furthermore, denoting $p_t(\cdot)$ as the marginal pdf of $Y_t$, its stationary distribution $p_t(Y_t) \propto \exp\left(-\frac{\|Y_t\|^2}{2t}\right)$, which converges to $\mathcal{N}(0, I)$ as $t \to 1$.*

We note that Lemma A.1 and Lemma A.2 follow from the duality between the heat equation and the continuity equation (Øksendal, 2003), where it is classically known that one can interpret a diffusive term as a vector field that is affine in the score function, and vice-versa. This connection has been carefully used to study a large family of stochastic interpolants (that generalize rectified flows) in (Albergo & Vanden-Eijnden, 2023; Albergo et al., 2023), and which can lead to a family of ODE-SDE pairs. In the lemmas above, we have provided explicit coefficients that have been directly derived, instead of using the stochastic interpolant formulation. Our key contribution lies in constructing a controlled ODEs (8) and (15), along with their equivalent SDEs (10) and (17) in Theorem 3.4 and Theorem 3.5, respectively. This aids faithfulness and editability as discussed in §4.

In **Lemma A.4**, we derive a rectified SDE that transforms noise into images by reversing the stochastic equivalent of rectified flows (12).

**Lemma A.4.** *Fix any small $\delta \in (0, 1)$, and for any $t \in [\delta, 1]$, the process $X_t$ governed by the SDE:*

$$\mathrm{d}X_t = \left[ \frac{1}{t}X_t + \frac{2(1-t)}{t}\nabla \log p_{1-t}(X_t) \right] \mathrm{d}t + \sqrt{\frac{2(1-t)}{t}}\mathrm{d}W_t, \quad X_0 \sim p_1, \tag{22}$$

*is the time-reversal of the SDE (12).*

**Implication.** The reverse SDE (22) provides a stochastic sampler for SoTA rectified flow models like Flux. Unlike diffusion-based generative models that explicitly model the score function $\nabla \log p_t(\cdot)$ in (22), rectified flows model a vector field, as discussed in §3.1. However, given a neural network $u(\mathbf{y}_t, t; \varphi))$ approximating the vector field $u_t(\mathbf{y}_t)$, **Lemma A.1** offers an explicit formula for computing the score function:

$$\nabla \log p_t(Y_t) = -\frac{1}{t}Y_t - \frac{1-t}{t}u(Y_t, t; \varphi). \tag{23}$$

This score function is used to compute the drift and diffusion coefficients of the SDE (22), resulting in a practically implementable stochastic sampler for Flux. This extends the applicability of Flux to downstream tasks where SDE-based samplers have demonstrated practical benefits, as seen in diffusion models (Ho et al., 2020; Song et al., 2021b; Rombach et al., 2022; Podell et al., 2023).

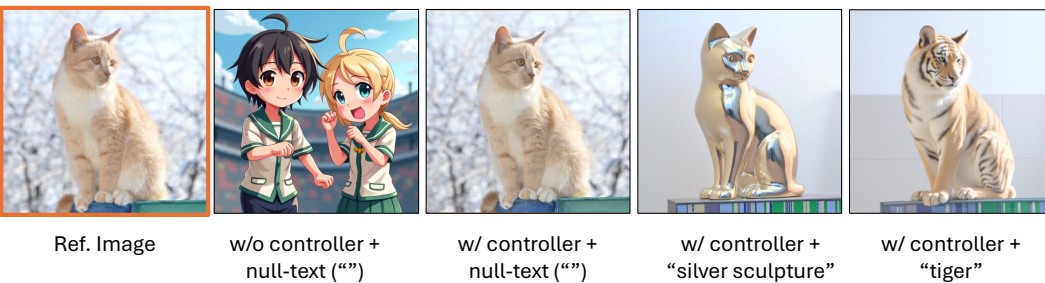

| Ref. Image | w/o controller + null-text ("") | w/ controller + null-text ("") | w/ controller + "silver sculpture" | w/ controller + "tiger" |

Figure 7: **Noise abduction and counterfactual sampling using our controlled rectified flows.** The first image is the original reference image of a cat. The second image is generated by mapping the original cat image to a structured noise and then transforming this noise into an image. In this case, we do not apply the controller and use null-text prompt during both inversion and generation. As expected, without the controller and text conditioning, the generated image appears unrelated to the original cat. Interestingly, when the controller is applied (and with null-text prompt) according to (15), the generated image closely reconstructs the original cat, illustrating that the structured noise preserves the salient features of the reference image, drawing a parallel to **noise abduction** discussed in (Pearl et al., 2016). The rightmost two images illustrate **counterfactual sampling**. Using the text prompts "A silver cat sculpture" and "A tiger", we generate images that correspond to the 'do' action in (Pearl et al., 2016). This demonstrates how our controlled rectified flow ODEs enable sampling of counterfactual images — imagining the original cat as a silver sculpture or a tiger.

## B    RF-INVERSION AS A COUNTERFACTUAL SAMPLER

RF-Inversion pipeline consists of three key steps: (i) Inversion – Mapping an image to a structured noise with null-text in RF vector field; (ii) Conditioning – Modifying the text input of the RF vector field to desired prompt; and (iii) Generation – Transforming the structured noise back to the desired image. This triplet can be interpreted as a prototype implementation of a counterfactual sampler. In particular, Pearl et al. (2016) (Section 4.2.4) propose a three-step process for counterfactual sampling in a Structural Causal Model (SCM): (i) Noise Abduction – Identifying the underlying noise sample that leads to the observed event; (ii) Action – Applying a "do" intervention on the SCM to encode new constraints; and (iii) Prediction (Sampling from the new SCM) – Using the abducted noise sample to generate outcomes in the updated SCM, ensuring sample-path coupling between the original and the modified SCMs.

As a concrete example, consider an image of a cat where we wish to generate silver sculpture of *the same* cat: see Figure 7. In the framework of Pearl et al. (2016), we start with an SCM as a world model. Given the observed event (the sample is the reference cat image), we first abduct to the underlying noise that led to this sample. This corresponds to the inversion step in our method, where we map the image to a structured noise using our controlled forward ODE (8). Next, we apply a "do" action by modifying the text conditioning of the RF model to "silver cat sculpture". In a flow-based model, text conditioning functions as an intervention because the score network is deterministic. This ensures meaningful propagation of the desired prompt through the generation process. Finally, we generate the new image using the controlled reverse ODE (15), producing a counterfactual version of the original cat — now imagined as a silver sculpture. Thus, RF Inversion has an interpretation as a counterfactual sampler working with a powerful world model (here, the combination of the Flux generative model (Black Forest Labs, 2024) and the optimal control flow)[1]. This interpretation, we believe, has several applications beyond image editing and stylization, and is a fruitful direction for future work.

---

[1]Without our controller, the base rectified flow model does not reconstruct the original image under a null-text input. Our controller is grounded on the original image, and hence enables the pipeline of (noise abduction, action, prediction). Other training-free approaches such as SDEdit (Meng et al., 2022) and DDIM Inversion (Song et al., 2021a) do not result in noise abduction because they do not provide an invertible and editable sample path coupling between the original image and noise, with both a null-text and text-edit prompt. As a result, they do not have this interpretation as a counterfactual sampler.

## C    TECHNICAL PROOFS

This section contains technical proofs of the theoretical results presented in this paper.

### C.1    PROOF OF PROPOSITION 3.2

*Proof.* The standard approach to solving an LQR problem is the minimum principle theorem that can be found in control literature (Fleming & Rishel, 1975; Basar et al., 2020). We follow this approach and provide the full proof below for completeness.

The Hamiltonian of the LQR problem (7) is given by

$$H(\mathbf{z}_t, \mathbf{p}_t, \mathbf{c}_t, t) = \frac{1}{2} \|\mathbf{c}_t\|^2 + \mathbf{p}_t^T \mathbf{c}_t. \tag{24}$$

For $\mathbf{c}_t^* = -\mathbf{p}_t$, the Hamiltonian attains its minumum value: $H(\mathbf{z}_t, \mathbf{p}_t, \mathbf{c}_t^*, t) = -\frac{1}{2} \|\mathbf{p}_t\|^2$. Using minimum principle theorem (Fleming & Rishel, 1975; Basar et al., 2020), we get

$$\frac{d\mathbf{p}_t}{dt} = \nabla_{\mathbf{z}_t} H(\mathbf{z}_t, \mathbf{p}_t, \mathbf{c}_t^*, t) = 0; \tag{25}$$

$$\frac{d\mathbf{z}_t}{dt} = \nabla_{\mathbf{p}_t} H(\mathbf{z}_t, \mathbf{p}_t, \mathbf{c}_t^*, t) = -\mathbf{p}_t; \tag{26}$$

$$\mathbf{z}_0 = \mathbf{y}_0; \tag{27}$$

$$\mathbf{p}_1 = \nabla_{\mathbf{z}_1} \left( \frac{\lambda}{2} \|\mathbf{z}_1 - \mathbf{y}_1\|_2^2 \right) = \lambda (\mathbf{z}_1 - \mathbf{y}_1). \tag{28}$$

From (25), we know $\mathbf{p}_t$ is a constant $\mathbf{p}$. Using this constant in (26) and integrating from $t \to 1$, we have $\mathbf{z}_1 = \mathbf{z}_t - \mathbf{p}(1 - t)$. Substituting $\mathbf{z}_1$ in (27),

$$\mathbf{p} = \lambda(\mathbf{z}_t - \mathbf{p}(1 - t) - \mathbf{y}_1) = \lambda(\mathbf{z}_t - \mathbf{y}_1) - \lambda(1 - t)\mathbf{p},$$

which simplifies to

$$\mathbf{p} = (1 + \lambda(1 - t))^{-1} \lambda(\mathbf{z}_t - \mathbf{y}_1) = \left( \frac{1}{\lambda} + (1 - t) \right)^{-1} (\mathbf{z}_t - \mathbf{y}_1).$$

Taking the limit $\lambda \to \infty$, we get $\mathbf{p} = \frac{\mathbf{z}_t - \mathbf{y}_1}{1 - t}$ and the optimal controller $\mathbf{c}_t^* = \frac{\mathbf{y}_1 - \mathbf{z}_t}{1 - t}$. Since $u_t(\mathbf{z}_t | \mathbf{y}_1) = \mathbf{y}_1 - \mathbf{y}_0$, the proof follows by substituting $\mathbf{y}_0 = \frac{\mathbf{z}_t - t\mathbf{y}_1}{1 - t}$. $\square$

### C.2    PROOF OF PROPOSITION 3.1

*Proof.* Initializing the generative ODE (1) with the structured noise $\mathbf{y}_1$, we get

$$\frac{dX_t}{dt} = v_t(X_t), \quad X_0 = \mathbf{y}_1, \quad \forall t \in [0, 1]. \tag{29}$$

Substituting $u_t(\cdot) = -v_{1-t}(\cdot)$ in ODE (6),

$$\frac{dY_t}{dt} = u_t(Y_t) = -v_{1-t}(Y_t), \quad Y_0 = \mathbf{y}_0, \quad \forall t \in [0, 1].$$

Replacing $t \to (1 - t)$,

$$\frac{dY_{1-t}}{dt} = v_t(Y_{1-t}), \quad \forall t \in [0, 1]. \tag{30}$$

Since (29) and (30) hold $\forall t \in [0, 1]$ and $X_0 = \mathbf{y}_1$, then $X_t = Y_{1-t}$ that implies $X_1 = Y_0 = \mathbf{y}_0$. $\square$

### C.3    PROOF OF THEOREM 3.4

*Proof.* From **Proposition 3.2**, we have $u_t(Y_t | Y_1) = \frac{Y_1 - Y_t}{1 - t}$. In **Lemma A.1**, we show that

$$u_t(Y_t) = \left[ -\frac{1}{1 - t} Y_t - \frac{t}{1 - t} \nabla \log p_t(Y_t) \right].$$

Now, the controlled ODE (8) becomes:

$$\mathrm{d}Y_t = \Big[u_t(Y_t) + \gamma\left(u_t(Y_t|Y_1) - u_t(Y_t)\right)\Big]\mathrm{d}t, \quad Y_0 \sim p_0, \quad Y_1 = \mathbf{y}_1$$

$$= \left[(1-\gamma)\left(-\frac{1}{1-t}Y_t - \frac{t}{1-t}\nabla\log p_t(Y_t)\right) + \gamma\left(\frac{Y_1 - Y_t}{1-t}\right)\right]\mathrm{d}t$$

$$= \left[-\frac{1}{1-t}Y_t - \frac{t}{1-t}(1-\gamma)\nabla\log p_t(Y_t) + \frac{\gamma}{1-t}Y_1\right]\mathrm{d}t$$

$$= \left[-\frac{1}{1-t}\left(Y_t - \gamma Y_1\right) - \frac{t}{1-t}(1-\gamma)\nabla\log p_t(Y_t)\right]\mathrm{d}t.$$

Using continuity equation (Øksendal, 2003), the density evolution of the controlled ODE (8) then becomes:

$$\frac{\partial p_{t,\gamma}(Y_t)}{\partial t} = \nabla\cdot\left[\left(\frac{1}{1-t}\left(Y_t - \gamma Y_1\right) + \frac{t}{1-t}(1-\gamma)\nabla\log p_t(Y_t)\right)p_{t,\gamma}(Y_t)\right]. \tag{31}$$

Applying Fokker-Planck equation (Øksendal, 2003) to the SDE (10), we have

$$\frac{\partial p_{t,\gamma}(Y_t)}{\partial t} + \nabla\cdot\left[\left(-\frac{1}{1-t}\left(Y_t - \gamma Y_1\right) - \frac{t}{1-t}(1-\gamma)\left(\nabla\log p_t(Y_t) - \nabla\log p_{t,\gamma}(Y_t)\right)\right)p_{t,\gamma}(Y_t)\right]$$
$$= \nabla\cdot\left[\frac{t}{1-t}(1-\gamma)\nabla p_{t,\gamma}(Y_t)\right],$$

which can be rearranged to equal (31) completing the proof. $\qquad\square$

## C.4 PROOF OF LEMMA A.1

*Proof.* Given $(Y_0, Y_1) \sim p_0 \times p_1$, the conditional flow matching loss (5) can be reparameterized as:

$$\mathcal{L}_{CFM}(\varphi) := \mathbb{E}_{t\sim\mathcal{U}[0,1],(Y_0,Y_1)\sim p_1\times p_0}\left[\|(Y_1 - Y_0) - u(Y_t, t; \varphi)\|_2^2\right], \quad Y_t = tY_1 + (1-t)Y_0,$$

where the optimal solution is given by the minimum mean squared estimator:

$$u_t(\mathbf{y}_t) = \mathbb{E}_{(Y_0,Y_1)\sim p_1\times p_0}\left[Y_1 - Y_0|Y_t = \mathbf{y}_t\right]. \tag{32}$$

Since $Y_t = tY_1 + (1-t)Y_0$, we use Tweedie's formula (Efron, 2011) to compute

$$\mathbb{E}\left[Y_0|Y_t = \mathbf{y}_t\right] = \frac{1}{1-t}\mathbf{y}_t + \frac{t^2}{1-t}\nabla\log p_t(\mathbf{y}_t). \tag{33}$$

Using the above relation, we obtain the following:

$$\mathbb{E}\left[Y_1|Y_t = \mathbf{y}_t\right] = \frac{1}{t}\mathbb{E}\left[Y_t - (1-t)Y_0|Y_t = \mathbf{y}_t\right]$$

$$= \frac{1}{t}\left(\mathbf{y}_t - (1-t)\mathbb{E}\left[Y_0|Y_t = \mathbf{y}_t\right]\right)$$

$$= \frac{1}{t}\left(\mathbf{y}_t - (1-t)\left(\frac{1}{1-t}\mathbf{y}_t + \frac{t^2}{1-t}\nabla\log p_t(\mathbf{y}_t)\right)\right)$$

$$= -t\,\nabla\log p_t(\mathbf{y}_t). \tag{34}$$

Combining (33) and (34) using linearity of expectation, we get

$$u_t(\mathbf{y}_t) = \mathbb{E}\left[Y_1|Y_t = \mathbf{y}_t\right] - \mathbb{E}\left[Y_0|Y_t = \mathbf{y}_t\right] \tag{35}$$

$$= -t\,\nabla\log p_t(\mathbf{y}_t) - \frac{1}{1-t}\mathbf{y}_t - \frac{t^2}{1-t}\nabla\log p_t(\mathbf{y}_t) \tag{36}$$

$$= -\frac{1}{1-t}\mathbf{y}_t - \frac{t}{1-t}\nabla\log p_t(\mathbf{y}_t), \tag{37}$$

The density evolution of $Y_t$ now immediately follows from the continuity equation (Øksendal, 2003) applied to (19). $\qquad\square$

## C.5 PROOF OF LEMMA A.2

*Proof.* The Fokker-Planck equation of the SDE (12) is given by

$$\frac{\partial p_t(Y_t)}{\partial t} + \nabla \cdot \left[ -\frac{1}{1-t} Y_t \, p_t(Y_t) \right] = \nabla \cdot \left[ \frac{t}{1-t} \, \nabla p_t(Y_t) \right]. \tag{38}$$

Rearranging (38) by multiplying and dividing $p_t(Y_t)$ in the right hand side, we get

$$\frac{\partial p_t(Y_t)}{\partial t} = \nabla \cdot \left[ \left( \frac{1}{1-t} Y_t + \frac{t}{1-t} \, \nabla \log p_t(Y_t) \right) p_t(Y_t) \right]. \tag{39}$$

To conclude, observe that that the density evolution above is identical to (20). □

## C.6 PROOF OF PROPOSITION A.3

*Proof.* The optimal vector field of the rectified flow ODE (6) is given by **Lemma A.1**. The proof then immediately follows from the Fokker-Planck equations in **Lemma A.1** and **Lemma A.2**.

From **Lemma A.2**, the density evolution of the SDE (12) is given by

$$\frac{\partial p_t(Y_t)}{\partial t} = \nabla \cdot \left[ \left( \frac{1}{1-t} Y_t + \frac{t}{1-t} \, \nabla \log p_t(Y_t) \right) p_t(Y_t) \right].$$

The stationary (or steady state) distribution satisfies the following:

$$\frac{\partial p_t(Y_t)}{\partial t} = 0 = \nabla \cdot \left[ \left( \frac{1}{1-t} Y_t + \frac{t}{1-t} \, \nabla \log p_t(Y_t) \right) p_t(Y_t) \right].$$

Using the boundary conditions (Øksendal, 2003), we get

$$\frac{1}{1-t} Y_t + \frac{t}{1-t} \, \nabla \log p_t(Y_t) = 0,$$

which immediately implies $p_t(Y_t) \propto e^{-\frac{\|Y_t\|^2}{2t}}$. □

## C.7 PROOF OF THEOREM 3.5

*Proof.* Using Fokker-Planck equation (Øksendal, 2003), **Lemma A.4** implies

$$\frac{\partial q_t(X_t)}{\partial t} = \nabla \cdot \left[ -q_t(X_t) \left( \frac{1}{t} X_t + \frac{1-t}{t} \nabla \log q_t(X_t) \right) \right].$$

Therefore, the optimal vector field $v_t(X_t)$ of the controlled ODE (15) is given by

$$v_t(X_t) = \frac{1}{t} X_t + \frac{1-t}{t} \nabla \log p_{1-t}(X_t). \tag{40}$$

The LQR problem (16) is identical to the LQR problem (7) with changes in the initial and terminal states. Similar to **Proposition 3.2**, we compute the closed-form solution for the conditional vector field of the ODE (15) as:

$$v_t(X_t|X_1) = \frac{X_1 - X_t}{1-t}. \tag{41}$$

Combining (40) and (41), we have

$$\begin{aligned}
\mathrm{d}X_t &= [v_t(X_t) + \eta(v_t(X_t|X_1) - v_t(X_t))] \, \mathrm{d}t \\
&= \left[ (1-\eta) \left( \frac{1}{t} X_t + \frac{1-t}{t} \nabla \log p_{1-t}(X_t) \right) + \eta \left( \frac{X_1 - X_t}{1-t} \right) \right] \mathrm{d}t \\
&= \left[ \frac{(1-\eta)(1-t) - \eta t}{t(1-t)} X_t + \frac{\eta}{1-t} X_1 + \frac{(1-\eta)(1-t)}{t} \nabla \log p_{1-t}(X_t) \right] \mathrm{d}t \\
&= \left[ \frac{1-t-\eta}{t(1-t)} X_t + \frac{\eta}{1-t} X_1 + \frac{(1-\eta)(1-t)}{t} \nabla \log p_{1-t}(X_t) \right] \mathrm{d}t.
\end{aligned}$$

The resulting continuity equation (Øksendal, 2003) becomes:

$$\frac{\partial q_{t,\eta}(X_t)}{\partial t} = \nabla \cdot \left[ -\left( \frac{1-t-\eta}{t(1-t)}X_t + \frac{\eta}{1-t}X_1 + \frac{(1-\eta)(1-t)}{t}\nabla \log p_{1-t}(X_t) \right) q_{t,\eta}(X_t) \right]$$

$$= \nabla \cdot \left[ -\left( \frac{1-t-\eta}{t(1-t)}X_t + \frac{\eta}{1-t}X_1 + \frac{(1-\eta)(1-t)}{t}\left(\nabla \log p_{1-t}(X_t) + \nabla \log q_{t,\eta}(X_t)\right) \right) q_{t,\eta}(X_t) \right.$$

$$\left. + \left( \frac{(1-\eta)(1-t)}{t}\nabla \log q_{t,\eta}(X_t) \right) q_{t,\eta}(X_t) \right].$$

Using time-reversal property from **Propsition 3.2**, the above expression simplifies to

$$\frac{\partial q_{t,\eta}(X_t)}{\partial t} + \nabla \cdot \left[ \left( \frac{1-t-\eta}{t(1-t)}X_t + \frac{\eta}{1-t}X_1 + \frac{(1-\eta)(1-t)}{t}\left(\nabla \log p_{1-t}(X_t) + \nabla \log p_{1-t,\eta}(X_t)\right) \right) q_{t,\eta}(X_t) \right]$$

$$= \nabla \cdot \left[ \frac{(1-\eta)(1-t)}{t}\nabla q_{t,\eta}(X_t) \right],$$

which yields the following SDE:

$$\mathrm{d}X_t = \left[ \frac{1-t-\eta}{t(1-t)}X_t + \frac{\eta}{1-t}X_1 + \frac{(1-\eta)(1-t)}{t}\left(\nabla \log p_{1-t}(X_t) + \nabla \log p_{1-t,\eta}(X_t)\right) \right] \mathrm{d}t$$

$$+ \sqrt{\frac{2(1-\eta)(1-t)}{t}}\mathrm{d}W_t,$$

and thus, completes the proof. $\square$

### C.8 Proof of Lemma A.4

*Proof.* It suffices to show that the Fokker-Planck equations of the SDE (22) and (12) are the same after time-reversal. Let $q_t(\cdot)$ denote the marginal pdf of $X_t$ such that $q_0(\cdot) = p_1(\cdot)$. The Fokker-Planck equations of the SDE (22) becomes

$$\frac{\partial q_t(X_t)}{\partial t} + \nabla \cdot \left[ q_t(X_t)\left( \frac{1}{t}X_t + \frac{2(1-t)}{t}\nabla \log p_{1-t}(X_t) \right) \right] = \nabla \cdot \left[ \left( \frac{1-t}{t} \right) \nabla q_t(X_t) \right],$$

which can be rearranged to give

$$\frac{\partial q_t(X_t)}{\partial t} = \nabla \cdot \left[ -q_t(X_t)\left( \frac{1}{t}X_t + \frac{2(1-t)}{t}\nabla \log p_{1-t}(X_t) \right) + \left( \frac{1-t}{t} \right) \nabla q_t(X_t) \right]$$

$$= \nabla \cdot \left[ -q_t(X_t)\left( \frac{1}{t}X_t + \frac{2(1-t)}{t}\nabla \log p_{1-t}(X_t) - \frac{1-t}{t}\nabla \log q_t(X_t) \right) \right]$$

Since $Y_t$ is the time-reversal process of $X_t$ as discussed in **Proposition (3.1)**,

$$\frac{\partial q_t(X_t)}{\partial t} = \nabla \cdot \left[ -q_t(X_t)\left( \frac{1}{t}X_t + \frac{1-t}{t}\nabla \log q_t(X_t) \right) \right].$$

Substituting $t \to 1-t$,

$$\frac{\partial q_{1-t}(X_{1-t})}{\partial t} = \nabla \cdot \left[ q_{1-t}(X_{1-t})\left( \frac{1}{1-t}X_{1-t} + \frac{t}{1-t}\nabla \log q_{1-t}(X_{1-t}) \right) \right],$$

which implies the density evolution of (12):

$$\frac{\partial p_t(Y_t)}{\partial t} = \nabla \cdot \left[ p_t(Y_t)\left( \frac{1}{1-t}Y_t + \frac{t}{1-t}\nabla \log p_t(Y_t) \right) \right].$$

This completes the proof of the statement. $\square$

# D    ADDITIONAL EXPERIMENTS

This section substantiates our contributions further by providing additional experimental details.

**Baselines.** We use the official NTI codebase[2] for the implementations of NTI (Mokady et al., 2023), P2P (Hertz et al., 2022), and DDIM (Song et al., 2021a) inversion. We use the official Diffusers implementation[3] for SDEdit and Flux[4]. We modify the pipelines for SDEdit and DDIM inversion to adapt to the Flux backbone. In Figure 6, for SDEdit, we use the optimally tuned strength parameter 0.7 (Meng et al., 2022), and for DDIM inversion, we use a midway starting point which is the same as ours ($s = 1 - 6/28 = 0.78$ for glasses and $s = 1 - 3/28 = 0.89$ for stroke2image).

For completeness, we include qualitative comparison with a leading training-based approach InstructPix2Pix (Brooks et al., 2023)[5] and a higher-order differential equation based LEDIT++ (Brack et al., 2024)[6] (§D). Table 3 summarizes the requirements of the compared baselines.

Table 3: Requirements of compared baselines. Our method outperforms prior works while requiring no additional training, optimization of prompt embedding, or attention manipulation scheme.

| Method | Training | Optimization | Attention Manipulation |
|---|---|---|---|
| SDEdit (Meng et al., 2022) | ✗ | ✗ | ✗ |
| DDIM (Song et al., 2021a) | ✗ | ✗ | ✗ |
| NTI (Mokady et al., 2023) | ✗ | ✓ | ✗ |
| NTI+P2P (Hertz et al., 2022) | ✗ | ✓ | ✓ |
| LEDIT++ (Brack et al., 2024) | ✗ | ✗ | ✓ |
| InstructPix2Pix (Brooks et al., 2023) | ✓ | ✗ | ✗ |
| Ours | ✗ | ✗ | ✗ |

**Metrics.** Following SDEdit (Meng et al., 2022), we measure faithfulness using L2 loss between the stroke input and the output image, and assess realism using Kernel Inception Distance (KID) between real and generated images. Stroke inputs are generated from RGB images using the algorithm provided in SDEdit. Given the subjective nature of image editing, we conduct a large-scale user study to calculate the user preference metric.

For face editing, we evaluate identity preservation, prompt alignment, and overall image quality using a face recognition metric (Ruiz et al., 2024), CLIP-T scores (Radford et al., 2021), and using CLIP-I scores (Radford et al., 2021), respectively. For the face recognition score, we calculate the L2 distance between the face embedding of the original image and the edited image, obtained from Inception ResNet trained on CASIA-Webface dataset. Similar to SDEdit (Meng et al., 2022), we conduct extensive experiments on Stroke2Image generation, and showcase additional capabilities qualitatively on a wide variety of semantic image editing tasks.

**Algorithm.** As shown in Figure 8, given a reference style or reference content denoted by $\mathbf{y}_0$, we first use **Algorithm 1** to obtain a structured noise $Y_1$ as discussed in §4. Then, we use **Algorithm 2** to transform the structured noise $Y_1$ back into an image based on a new text prompt.

## D.1    HYPER-PARAMETER CONFIGURATIONS

In Table 4, we provide the hyper-parameters for the empirical results reported in §5. We use a fix $\gamma = 0.5$ in our controlled forward ODE (8) and a time-varying guidance parameter $\eta_t$ in our controlled reverse ODE (15), as motivated in **Remark 3.3** and **Remark 3.6**. Thus, our algorithm introduces one additional hyper-parameter $\eta_t$ into the Flux pipeline. For each experiment, we use a fixed time-varying schedule of $\eta_t$ described by starting time ($s$), stopping time $\tau$, and strength ($\eta$).

---

[2] https://github.com/google/prompt-to-prompt
[3] https://github.com/huggingface/diffusers
[4] https://github.com/black-forest-labs/flux
[5] https://huggingface.co/spaces/timbrooks/instruct-pix2pix
[6] https://huggingface.co/spaces/editing-images/leditsplusplus

---

**Algorithm 1:** Controlled Forward ODE (8)

---

**Input:** Discretization steps $N$, reference image $\mathbf{y}_0$, prompt embedding network $\Phi$, Flux model
$\quad\quad u(\cdot,\cdot,\cdot;\varphi)$, Flux noise scheduler $\sigma : [0,1] \rightarrow \mathbb{R}$

**Tunable parameter:** Controller guidance $\gamma$

**Output:** Structured noise $Y_1$

1 Initialize $Y_0 = \mathbf{y}_0$
2 Fix a noise sample $\mathbf{y}_1$
3 **for** $i = 0$ **to** $N-1$ **do**
4 $\quad$ Current time step: $t_i = \frac{i}{N}$
5 $\quad$ Next time step: $t_{i+1} = \frac{i+1}{N}$
6 $\quad$ Unconditional vector field: $u_{t_i}(Y_{t_i}) = u(Y_{t_i}, t_i, \Phi(\text{""}); \varphi)$ $\quad\quad\quad$ ▷ **Proposition** 3.1
7 $\quad$ Conditional vector field: $u_{t_i}(Y_{t_i}|\mathbf{y}_1) = \frac{\mathbf{y}_1 - Y_{t_i}}{1 - t_i}$ $\quad\quad\quad$ ▷ **Proposition** 3.2
8 $\quad$ Controlled vector field: $\hat{u}_{t_i}(Y_{t_i}) = u_{t_i}(Y_{t_i}) + \gamma \left( u_{t_i}(Y_{t_i}|\mathbf{y}_1) - u_{t_i}(Y_{t_i}) \right)$ $\quad$ ▷ODE (8)
9 $\quad$ Next state: $Y_{t_{i+1}} = Y_{t_i} + \hat{u}_{t_i}(Y_{t_i})\left(\sigma(t_{i+1}) - \sigma(t_i)\right)$
10 **end**
11 **return** $Y_1$

---

**Algorithm 2:** Controlled Reverse ODE (15)

---

**Input:** Discretization steps $N$, reference text "prompt", reference image $\mathbf{y}_0$, prompt embedding
$\quad\quad$ network $\Phi$, Flux model $u(\cdot,\cdot,\cdot;\varphi)$, Flux noise scheduler $\sigma : [0,1] \rightarrow \mathbb{R}$,
$\quad\quad$ structured noise $\mathbf{y}_1$

**Tunable parameter:** Controller guidance $\eta$

**Output:** Edited image $X_1$

1 Initialize $X_0 = \mathbf{y}_1$
2 **for** $i = 0$ **to** $N-1$ **do**
3 $\quad$ Current time step: $t_i = \frac{i}{N}$
4 $\quad$ Next time step: $t_{i+1} = \frac{i+1}{N}$
5 $\quad$ Unconditional vector field: $v_{t_i}(X_{t_i}) = -u(X_{t_i}, 1 - t_i, \Phi(\text{prompt}); \varphi)$ $\quad$ ▷ **Proposition** 3.1
6 $\quad$ Conditional vector field: $v_{t_i}(X_{t_i}|\mathbf{y}_0) = \frac{\mathbf{y}_0 - X_{t_i}}{1 - t_i}$ $\quad\quad\quad$ ▷ **Proposition** 3.2
7 $\quad$ Controlled vector field: $\hat{v}_{t_i}(X_{t_i}) = v_{t_i}(X_{t_i}) + \eta \left( v_{t_i}(X_{t_i}|\mathbf{y}_0) - v_{t_i}(X_{t_i}) \right)$ $\quad$ ▷ODE (15)
8 $\quad$ Next state: $X_{t_{i+1}} = X_{t_i} + \hat{v}_{t_i}(X_{t_i})\left(\sigma(t_{i+1}) - \sigma(t_i)\right)$
9 **end**
10 **return** $X_1$

---

We use the default config for Flux model: 3.5 for classifier-free guidance and 28 for the total number of inference steps.

Table 4: Hyper-parameter configuration of our method for inversion and editing tasks.

| Task | Starting Time ($s$) | Controller Guidance ($\eta_t$) | |
| --- | --- | --- | --- |
| | | Stopping Time ($\tau$) | Strength ($\eta$) |
| Stroke2Image | 3 | 5 | 0.9 |
| Object insert | 0 | 6 | 1.0 |
| Gender editing | 0 | 8 | 1.0 |
| Age editing | 0 | 5 | 1.0 |
| Adding glasses | 6 | 25 | 0.7 |
| Stylization | 0 | 6 | 0.9 |
| Inversion only | 8 | 25 | 1 |

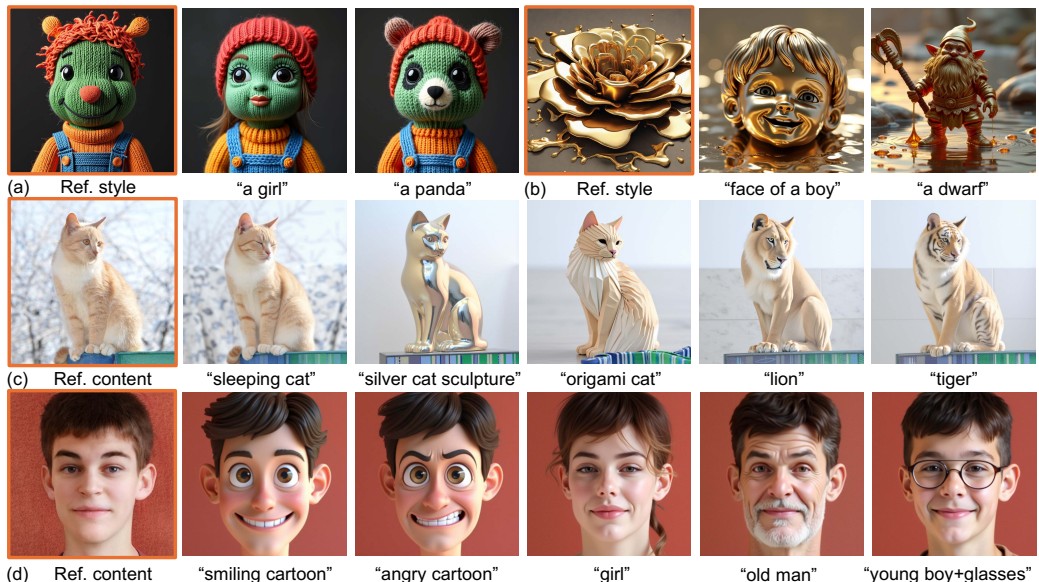

Figure 8: **Rectified flows for image inversion and editing.** Our approach efficiently inverts reference style images in (a) and (b) without requiring text descriptions of the images and applies desired edits based on new prompts (e.g. "a girl" or "a dwarf"). For a reference content image (e.g. a cat in (c) or a face in (d)), it performs semantic image editing (e.g. "sleeping cat") and stylization (e.g. "a photo of a cat in origmai style"), without leaking unwanted content from the reference image. Input images have orange borders.

## D.2 ABLATION STUDY

In this section, we conduct ablation study for our controller guidance parameter $\eta_t$. We consider two different time-varying schedules for $\eta_t$, and show that our controller strength allows for a smooth interpolation between unconditional and conditional generation.

In Figure 9, we show the effect of starting time in controlling the faithfulness of inversion; starting time $s \in [0, 1]$ is defined as the time at which our controlled reverse ODE (15) is initialized. The initial state $X_s = \mathbf{y}_{1-s}$ is obtained by integrating the controlled forward ODE (8) from $0 \to 1 - s$.

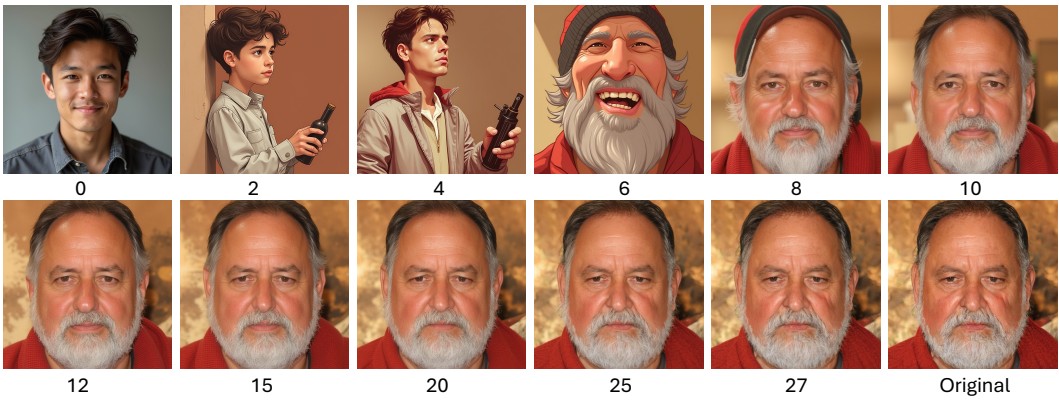

Figure 9: **Effect of starting time.** Prompt: "A young man". The number below each figure denotes the starting time scaled by 28 (the total number of denoising steps) for better interpretation. In the absence of controller guidance ($\eta_t = 0$), increasing the starting time ($s$) in our controlled ODE (15) improves faithfulness to the original image.

In Figure 10, we study the effect of stopping time. We find that increasing controller guidance $\eta_t$ by increasing the stopping time $\tau$ guides the reverse flow towards the original image. However, we observe a phase transition around $\tau = 0.14 = 4/28$, indicating that the resulting drift in our controlled reverse ODE (15) is dominated by the conditional vector field $v_t(X_t|\mathbf{y}_0)$ for $t \geq \tau$. Therefore, the reverse flow solves the LQR problem (16) and drives toward the terminal state (i.e., the original image).

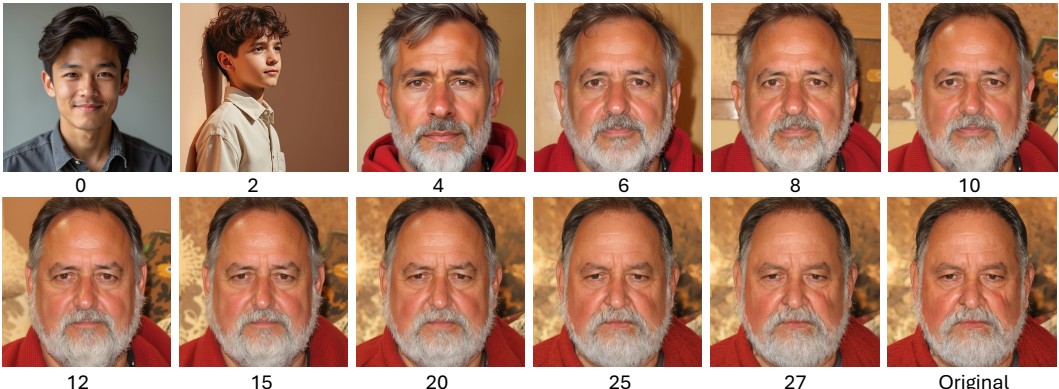

Figure 10: **Effect of controller guidance.** Prompt: "A young man". For a fixed starting time $s = 0$, consider a time-varying controller guidance schedule $\eta_t = \eta \; \forall t \leq \tau$ and 0 otherwise. The number below each figure denotes the stopping time $\tau$ scaled by 28 (the total number of denoising steps) for better interpretation. Increasing $\tau$ increases the controller guidance ($\eta_t$) that improves faithfulness to the original image.

In Figure 11, we visualize the effect of our controller guidance for another time-varying schedule. We make a similar observation as in Figure 10: increasing $\eta_t$ improves faithfulness. However, we notice a smooth transition from the unconditional to the conditional vector field, evidence from the smooth interpolation between "A young man" at the top left ($\eta = 0$) and the original image at the bottom right.

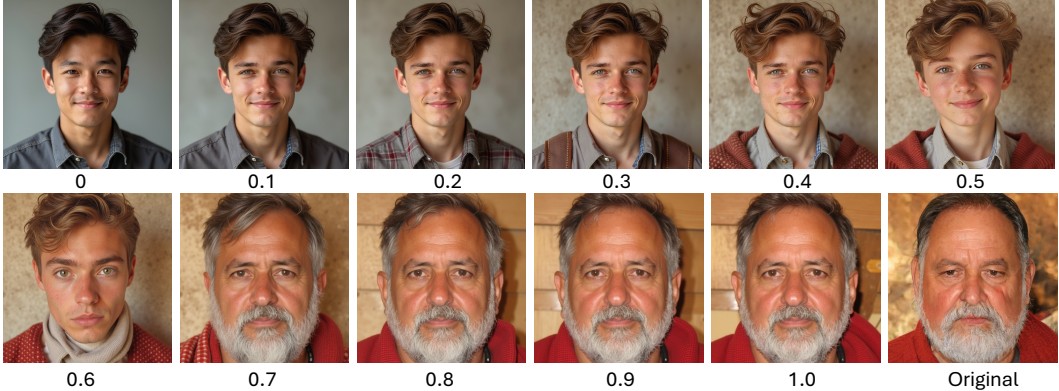

Figure 11: **Effect of controller guidance** for another time-varying schedule. Prompt: "A young man". For a fixed starting time $s = 0$ and stopping time $\tau = 8/28$, consider a time-varying controller guidance schedule $\eta_t = \eta \; \forall t \leq \tau$ and 0 otherwise. Increasing $\eta$ increases the controller guidance ($\eta_t$) that improves faithfulness to the original image.

### D.3 NUMERICAL SIMULATION

In this section, we design synthetic experiments to compare reconstruction accuracy of DM and RF inversion. Given $Y_0 \sim p_0$, where the data distribution $p_0 \coloneqq \mathcal{N}(\mu, I)$ and the source distribution

$q_0 := \mathcal{N}(0, I)$, we numerically simulate the ODEs and SDEs associated with DM and RF inversion; see our discussion in §3.

For $\mu = 10$, we fix $\gamma = 0.5$ in the controlled forward ODE (8), and $\eta = 0.5$ in the controlled reverse ODE (15). These ODEs are simulated using the Euler discretization scheme with 100 steps. Additionally, we simulate the uncontrolled rectified flow ODEs (6) $\rightarrow$ (1) as a special case of our controlled ODEs (8) $\rightarrow$ (15) by setting $\gamma = \eta = 0$, and the deterministic diffusion model DDIM (Song et al., 2021a) in the same experimental setup.

The inversion accuracy is reported in Table 5. Observe that RF inversion has less L2 and L1 error compared to DDIM inversion (14). The minimum error is obtained by setting $\gamma = \eta = 0$ (i.e., reversing the standard rectified flows), which supports our discussion in §3.3.

Furthermore, we simulate the stochastic samplers corresponding to these ODEs in Table 5, highlighted in orange. Similar to the deterministic samplers, we observe that stochastic equivalents of rectified flows more accurately recover the original sample compared to diffusion models. Our controller in RF Inversion (10) $\rightarrow$ (17) effectively reduces the reconstruction error in the uncontrolled RF Inversion (12) $\rightarrow$ (22), which are special cases when $\gamma = \eta = 0$. Thus, we demonstrate that (controlled) rectified stochastic processes are better at inverting a given sample from the target distribution, outperforming the typical OU process used in diffusion models (Song & Ermon, 2019; Ho et al., 2020; Song et al., 2021a;b).

Table 5: DM and RF inversion accuracy. Stochastic samplers are highlighted in orange.

| Method | L2 Error | L1 Error |
|---|---|---|
| DDIM Inversion (14) | 6.024 | 19.038 |
| DDPM Inversion (13) | 6.007 | 15.758 |
| RF Inversion ($\gamma = \eta = 0$) (8) $\rightarrow$ (15) | 0.092 | 0.20 |
| RF Inversion ($\gamma = \eta = 0$) (10) $\rightarrow$ (17) | 3.564 | 8.795 |
| RF Inversion ($\gamma = 0.5, \eta = 0$) (8) $\rightarrow$ (15) | 4.777 | 11.628 |
| RF Inversion ($\gamma = 0, \eta = 0.5$) (8) $\rightarrow$ (15) | 1.219 | 3.074 |
| RF Inversion ($\gamma = 0.5, \eta = 0.5$) (8) $\rightarrow$ (15) | 0.628 | 1.643 |
| RF Inversion ($\gamma = \eta = 0.5$) (10) $\rightarrow$ (17) | 0.269 | 0.694 |
| RF Inversion ($\gamma = \eta = 1.0$) (10) $\rightarrow$ (17) | 0.003 | 0.010 |

In Figure 12, we compare sample paths of diffusion models and recitified flows using 10 IID samples drawn from $p_0$. In Figure 13, we visualize paths for those samples using our controlled ODEs and SDEs with $\gamma = \eta = 0.5$.

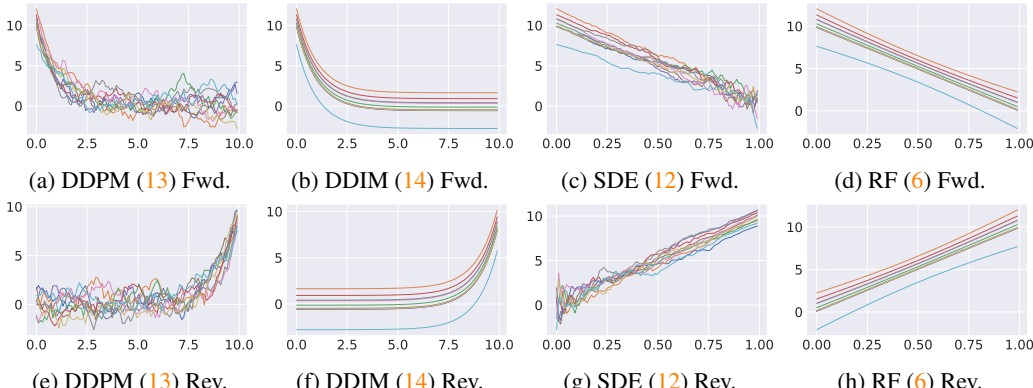

|  |  |  |  |
|---|---|---|---|
| (a) DDPM (13) Fwd. | (b) DDIM (14) Fwd. | (c) SDE (12) Fwd. | (d) RF (6) Fwd. |
| (e) DDPM (13) Rev. | (f) DDIM (14) Rev. | (g) SDE (12) Rev. | (h) RF (6) Rev. |

Figure 12: **Sample paths of DMs and RFs.** Top row corresponds to the forward process $\{Y_t\}$, and bottom row, reverse process $\{X_t\}$. In each plot, time is along the horizontal axis and the process, along the vertical axis. The sample paths of RFs are straighter than that of DMs, allowing coarse discretization and faithful reconstruction.

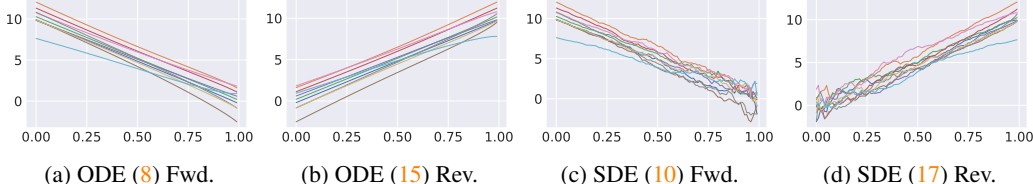

(a) ODE (8) Fwd.     (b) ODE (15) Rev.     (c) SDE (10) Fwd.     (d) SDE (17) Rev.

Figure 13: **Sample paths of our controlled ODEs and SDEs.** (a,c) The optimal controller $u_t(Y_t|Y_1)$ steers $Y_t$ towards the terminal state $Y_1 \sim p_1$ during inversion. (b,d) Similarly, $v_t(X_t|Y_0)$ guides $X_t$ towards the reference image $Y_0 \sim p_0$, significantly reducing the reconstruction error.

## D.4   ADDITIONAL RESULTS ON STROKE2IMAGE GENERATION

In Figure 14 and Figure 15, we show additional qualitative results on Stroke2Image generation. Our method generates more realistic images compared to leading training-free approaches in semantic image editing including optimization-based NTI (Mokady et al., 2023) and attention-based NTI+P2P (Hertz et al., 2022). Furthermore, it gives a competitive advantage over the training-based approach InstructPix2Pix (Brooks et al., 2023).

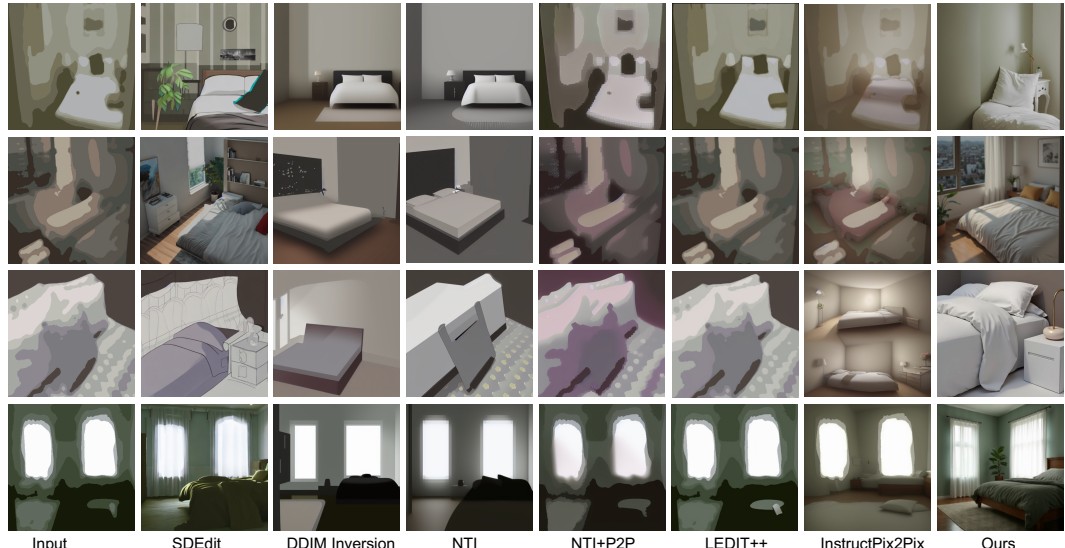

Input     SDEdit     DDIM Inversion     NTI     NTI+P2P     LEDIT++     InstructPix2Pix     Ours

Figure 14: **Stroke2Image generation.** Additional qualitative results on LSUN-Bedroom dataset comparing our method with SoTA training-free and training-based editing approaches.

In Figure 16, we demonstrate the robustness of our approach to corruption at initialization. All the methods transform the stroke input (corrupt image) to a structured noise, which is again transformed back to a similar looking stroke input, highlighting the faithfulness of these methods. However, unlike our approach, the resulting images in other methods are not editable given a new prompt.

## D.5   ADDITIONAL RESULTS ON SEMANTIC IMAGE EDITING

Figure 17 illustrates a smooth interpolation between "A man" → "A woman" (top row) and "A woman" → "A man" (bottom row). The facial expression and the hair style are gradually morphed from one person to the other.

In Figure 18, we show the ability to regulate the extent of age editing. Given an image of a young woman and the prompt "An old woman", we gradually reduce the controller strength $\eta_t$ to make the person look older. Similarly, we reduce the strength to make an old man look younger.

Figure 19 shows the insertion of multiple objects by text prompts, such as "pepperoni", "mushroom", and "black leaves" to an image of a pizza. Interestingly, pepperoni is not deleted while

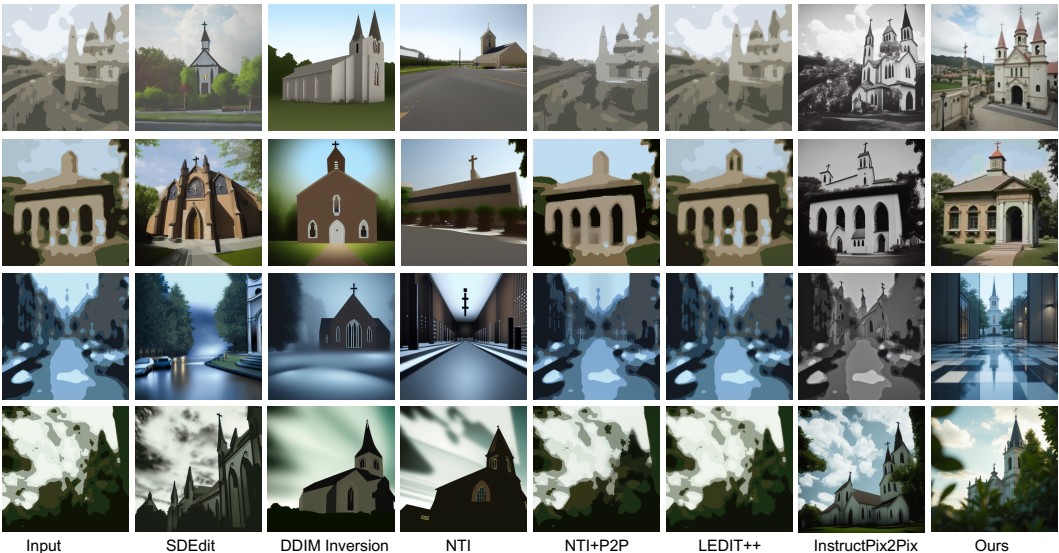

Figure 15: **Stroke2Image generation.** Additional qualitative results on LSUN-Church dataset comparing our method with SoTA training-free and training-based editing approaches.

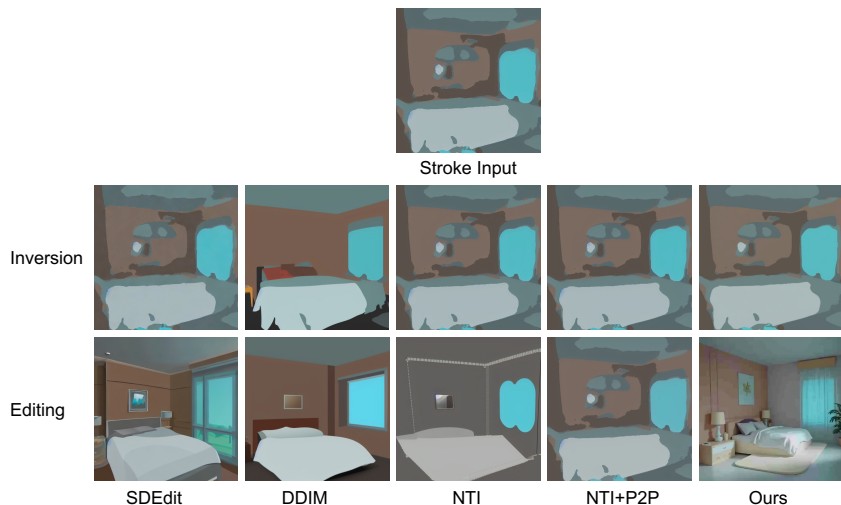

Figure 16: **Robustness.** For inversion, all methods perform well at recovering the stroke input when given a null prompt. However, when a new prompt like "a photo-realistic picture of a bedroom" is provided, only our method successfully generates realistic images. The other methods continue to suffer from the initial corruption, failing to make the output more realistic.

inserting mushroom, and mushroom is not deleted while inserting black leaves. The product is finally presented in a lego style.

Figure 20 captures a variety of facial expressions that stylize a reference image. Given the original image and text prompt: e.g. "Face of a girl in disney 3d cartoon style", we first invert the image to generate the stylized version of the original image. Then, we add the prompt for the expression (e.g., "surprised") at the end of the prompt and run our editing algorithm (15) with this new prompt: "Face of a girl in disney 3d cartoon style, surprised". By changing the expression, we are able to preserve the identity of the stylized girl and generate prompt-based facial expressions.

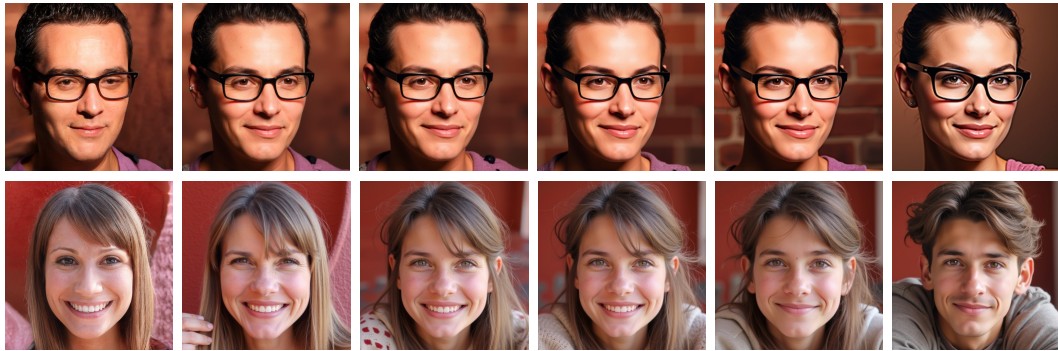

Figure 17: **Gender editing.** Our method smoothly interpolates between "A man" ↔ "A woman".

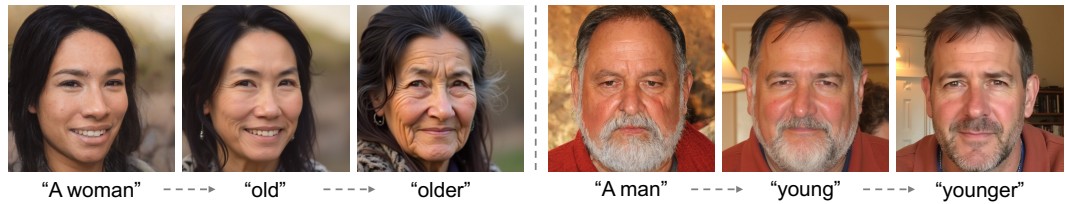

Figure 18: **Age editing.** Our method regulates the extent of age editing.

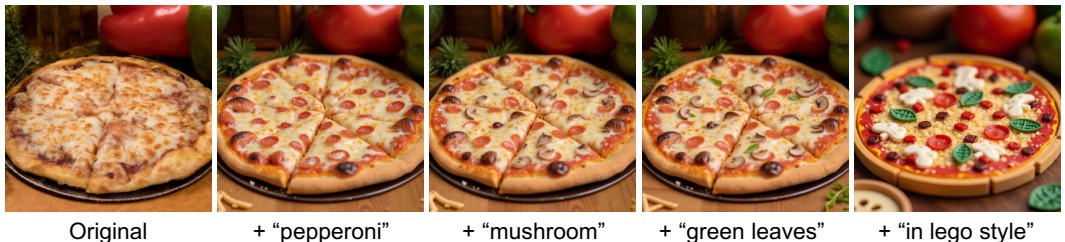

Figure 19: **Object insert.** Text-guided insertion of multiple objects sequentially.

## D.6 HUMAN EVALUATION

We conduct a user study on the test splits of both LSUN Bedroom and LSUN Church dataset using Amazon Mechanical Turk, with 126 participants in total. As shown in Figure 21, each question was accompanied by an explanation of the task, the question, and the evaluation criteria. Participants were shown a pair of stroke-to-image outputs from different models, in random order, along with the input stroke image. They were asked to select one of three options based on their preference using the following two criteria:

1. **Realism:** which of these two images look more like a real, photorealistic image?
2. **Faithfulness:** which of these two images match more closely to the input stroke image?

We collect 3 responses per question. With 300 images in the test dataset and 10 pairwise comparisons, we gathered 9,000 responses for this evaluation. The example in Figure 21 is for the LSUN Church dataset; for LSUN Bedroom dataset, we simply replace the word "church" to "bedroom" in the instructions.

## D.7 GENERATIVE MODELING USING RECTIFIED STOCHASTIC DIFFERENTIAL EQUATIONS

In Figure 22, we compare images generated by the ODE (1) and SDE (22) variant of Flux across different discretization steps. Figure 23 illustrates text-to-image generation using the stochastic

"Face of a young girl in disney 3d cartoon style"

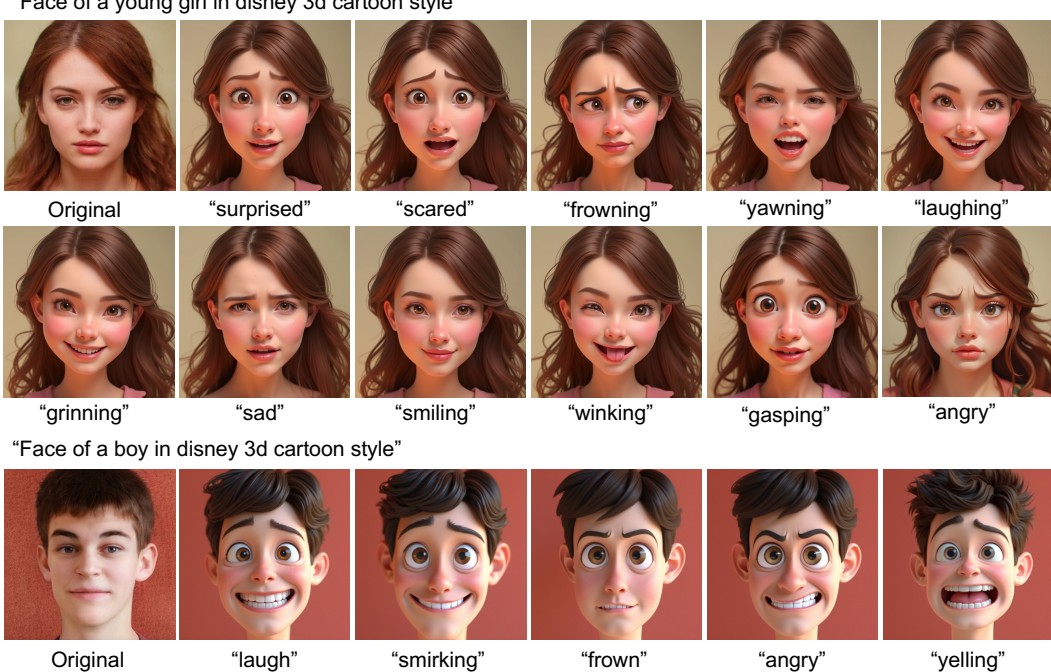

"Face of a boy in disney 3d cartoon style"

Figure 20: **Stylization of a reference image given prompt-based facial expressions.** Given the reference content (e.g., an image of a woman) and the desired prompt (e.g., "a young girl in disney 3d cartoon style"), our method makes the person younger and follows the disney 3d cartoon style. This demonstrates the ability of our method to generate various stylized expressions.

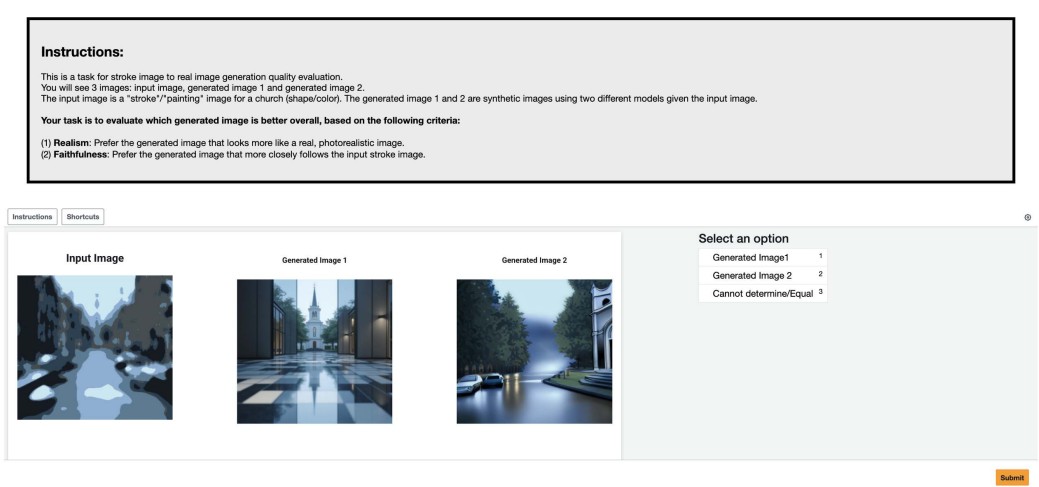

Figure 21: **Interface for human evaluation.** Each participant is asked to select their preferred image based on two criteria: *realism* and *faithfulness*.

sampler for Flux[7], highlighting the practical significance of our theoretical findings in §3 and Appendix A.

## D.8 DISCUSSION ON ACCELERATING CONTROLLED RECTIFIED FLOWS

---

[7]https://github.com/black-forest-labs/flux

FluxODE

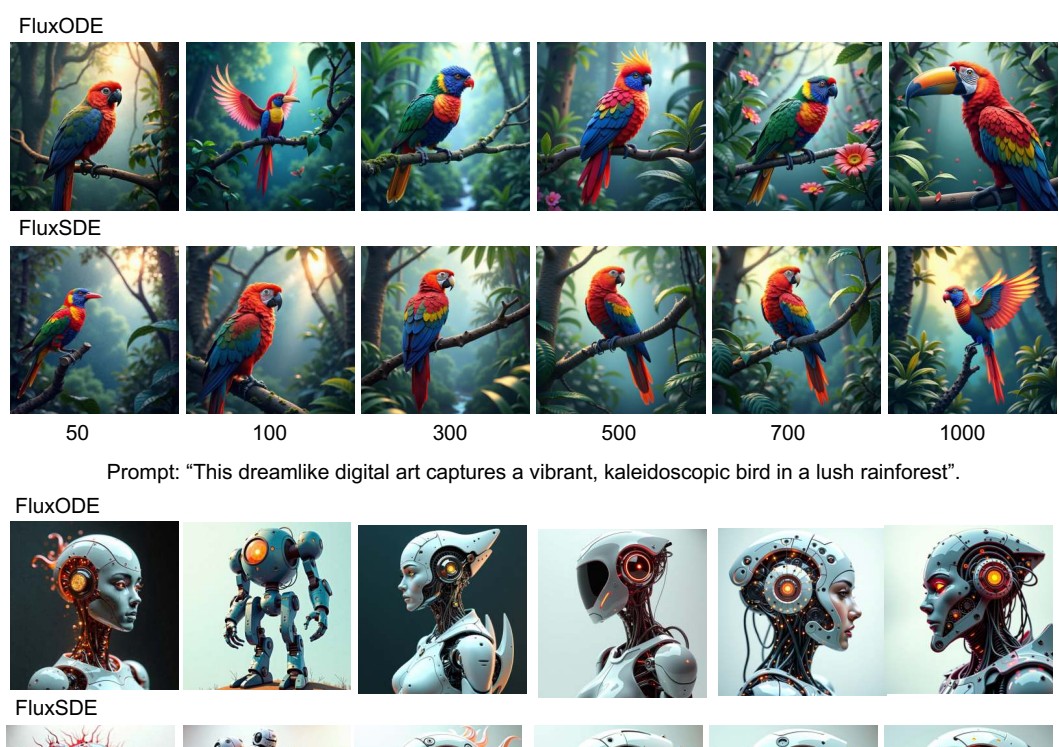

FluxSDE

50    100    300    500    700    1000

Prompt: "This dreamlike digital art captures a vibrant, kaleidoscopic bird in a lush rainforest".

FluxODE

FluxSDE

50    100    300    500    700    1000

Prompt: "A futuristic robot depicted entirely out of fractals".

Figure 22: **T2I generation** using rectified ODE (top) and SDE (bottom) for different number of discretization steps marked along the X-axis. The stochastic equivalent sampler FluxSDE generates samples visually comparable to FluxODE at different levels of discretization.

In this paper, we used 28 inference steps, the default setting for Flux. It is well established that reconstruction error increases as the number of inference steps decreases due to coarse discretization. Prior works on diffusion models have developed methods to mitigate this error with fewer steps (Garibi et al., 2024; Pan et al., 2023). In particular, the accelerated iteration procedure from AIDI (Pan et al., 2023) and the renoising iteration step in ReNoise (Garibi et al., 2024) could enhance the inversion and editing capabilities of our method with fewer steps. Integration of such sampling techniques in rectified flows represents a promising direction for future work.

## D.9 RF-INVERSION AS A PLUG-AND-PLAY SOLUTION FOR RECTIFIED FLOWS

Figure 24 presents stylization results achieved using RF-Inversion with Stable Diffusion 3.5 (Esser et al., 2024). The generated images accurately reflect the artistic styles of the reference style images while faithfully adhering to the desired prompts. The top row includes reference styles from StyleAligned (Hertz et al., 2023) benchmark. The bottom row contains hand-drawn styles: plastic crayon[8] and pencil sketch[9].

## D.10 RF-INVERSION WITH DIFFERENT INITIALIZATION

---

[8]https://www.pinterest.com/pin/the-sunset-drawing-ideas-6-with-plastic-crayons-cool-web-fun
[9]https://www.pinterest.com/pin/378020962481314381/

| **Flux** | **FluxSDE (Ours)** | **Flux** | **FluxSDE (Ours)** |
|---|---|---|---|

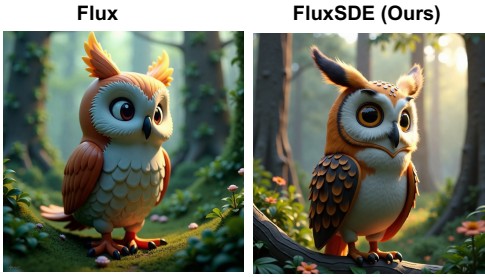 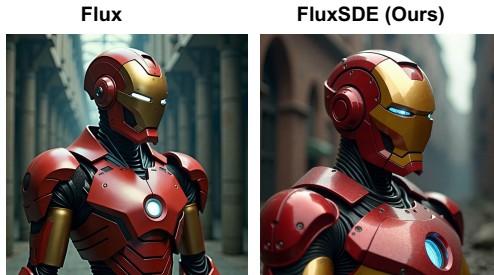

Prompt: "portrait, looking to one side of frame, lucid dream-like 3d model of an owl, video game character, forest, wonderland, photorealism, cinematic artistic style."

Prompt: "a robot with a reflective helmet, iron armor, photorealistic, in shades of red and golden brown, dark gloomy environment, epic scene."

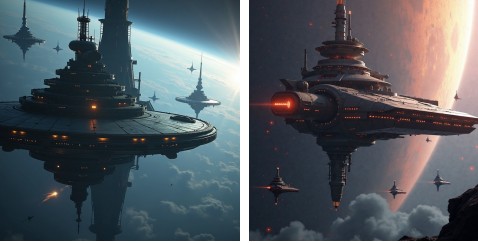 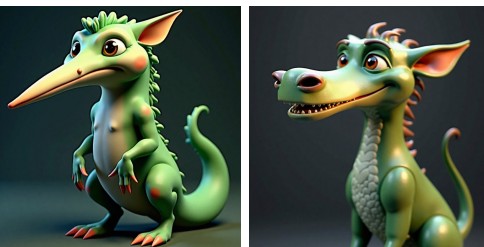

Prompt: "a space elevator, cinematic scifi art, spacecrafts flying in the background"

Prompt: "a 3d model of a magical creature, long nose, green color, movie asset, ultra detailed."

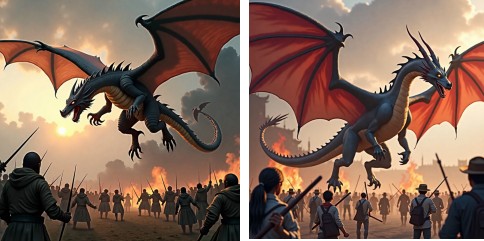 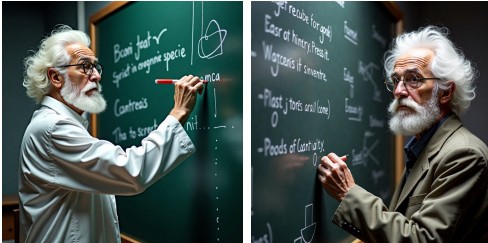

Prompt: "a dragon soaring through the sky, battle ground, people fighting on the ground."

Prompt: "a genius scientist, in his 60s stands, writing on the black board, white hair, white beard, round spectacles."

Figure 23: **T2I generation using rectified flow SDE (22).** Our stochastic sampler is visually comparable to the standard deterministic sampler provided by Flux.

Figure 25 shows the effect of $\mathbf{y}_1$ on reference image-based stylization and structure preservation. The proposed method reliably generates accurately stylized image and preserves the structure of the input image. While the choice of $\mathbf{y}_1$ does have an impact, our controller ensures that the semantics of the reference input are still preserved for different random choices of $\mathbf{y}_1$.

Regarding stroke-to-image quality, the manually annotated clean stroke images used in SDEdit are not publicly available. Therefore, we simulate stroke images following Section D.2 (Figures 30 and 31) in SDEdit, as this strategy was also used in their large-scale evaluation. The simulated stroke images are noisy which increases variability in the layout of the generated images. We also added one example in Figure 25 (d) using an annotated bedroom stroke input (by taking a screenshot of stroke paint from Figure 1 of SDEdit), showing that the layout of the generated images are better aligned with a cleaner stroke input.

## D.11 COMPATIBILITY OF RF-INVERSION WITH FLUX-LORA

Figure 26 illustrates content-style composition using Flux-LoRA (Hu et al., 2021). In this experiment, we apply LoRA fine-tuning only for content while using our method for stylization. We train two LoRA models with the $<$ sks $>$ token using images from the DreamBooth (Ruiz et al., 2023)

dataset[10]: (1) "a sks dog" (6 images) and (2) "a sks cat" (5 images). For each LoRA, we use rank 16 and fine-tune for 1000 iterations, which takes around 25 minutes. We first use $u_t(\mathbf{y}_t)$ in **Algorithm 1** to obtain a structured noise corresponding to the reference style image (e.g., "line drawing"). Subsequently, we employ **Algorithm 2** with LoRA weights added to $v_t(\cdot)$ using the desired prompt (e.g., "a sks dog in line drawing style"). Observe that the unique identifiers of the dog (e.g., the beard) are effectively captured in the generated sample.

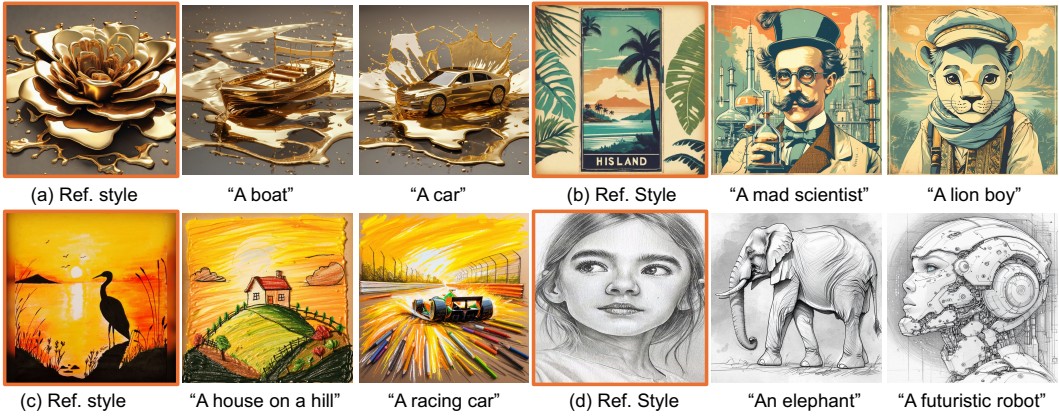

(a) Ref. style      "A boat"      "A car"      (b) Ref. Style      "A mad scientist"      "A lion boy"

(c) Ref. style      "A house on a hill"      "A racing car"      (d) Ref. Style      "An elephant"      "A futuristic robot"

Figure 24: **Compatibility of our method with another RF model, Stable Diffusion 3.5.** Reference styles in (a) "melting golden 3d rendering" and (b) "vintage travel poster" are from StyleAligned benchmark (Hertz et al., 2023), and (c) "plastic crayon drawing art" and (d) "pencil sketch" are hand-drawn styles provided by users. Please see §D.10 for the reference style credits.

## D.12 ADAPTABILITY OF RF-INVERSION TO GENERAL INVERSE PROBLEMS

Figure 27 shows that our method RF-Inversion easily extends to a broad class of inverse problems without using additional training, latent variable optimization, or complex attention processors. For restoration task, we consider super-resolution by 8X (left) and motion blur by a kernel $(61 \times 61)$(right).

## D.13 STUDY OF CONTROLLER GUIDANCE IN IMAGE EDITING

Figure 28 demonstrates the influence of the controller guidance parameters $\gamma$ (forward flow) and $\eta$ (reverse flow) on the image transformation process. The interplay between $\gamma$ and $\eta$ is crucial for balancing the forward and reverse flows, ensuring that the resulting output follows the structure of the given input. Increasing $\gamma$ improves realism but the structure changes. For instance, the church door is facing towards right. For a fixed $\gamma$ (say 0.5), increasing $\eta$ aligns the structure with the reference input ($\mathbf{y}_0$). Interestingly, at a midway point ($\gamma = 0.5, \eta = 0.5$), the image looks blurry because this is a superposition of two conflicting images (e.g., in one image, the door is facing to the right, and in another, to the front). Finally, our controller guidance ($\eta$) rectifies this process to become more faithful to the reference input.

## D.14 ADDITIONAL EDITING RESULTS USING 8 STEP DISTILLED MODEL

Figure 29 shows additional qualitative results on face editing and stroke2image generation using our method integrated with Flux-Turbo-LoRA[11]. These results demonstrate the compatibility of our method with a distilled base model capable of sampling in as few as 8 steps. In this experiment, we employ the distilled Flux-Turbo-LoRA model for computing $u_t(\cdot)$ and $v_t(\cdot)$ in our **Algorithm 1** and **Algorithm 2**, respectively.

---

[10] https://github.com/google/dreambooth
[11] https://huggingface.co/alimama-creative/FLUX.1-Turbo-Alpha

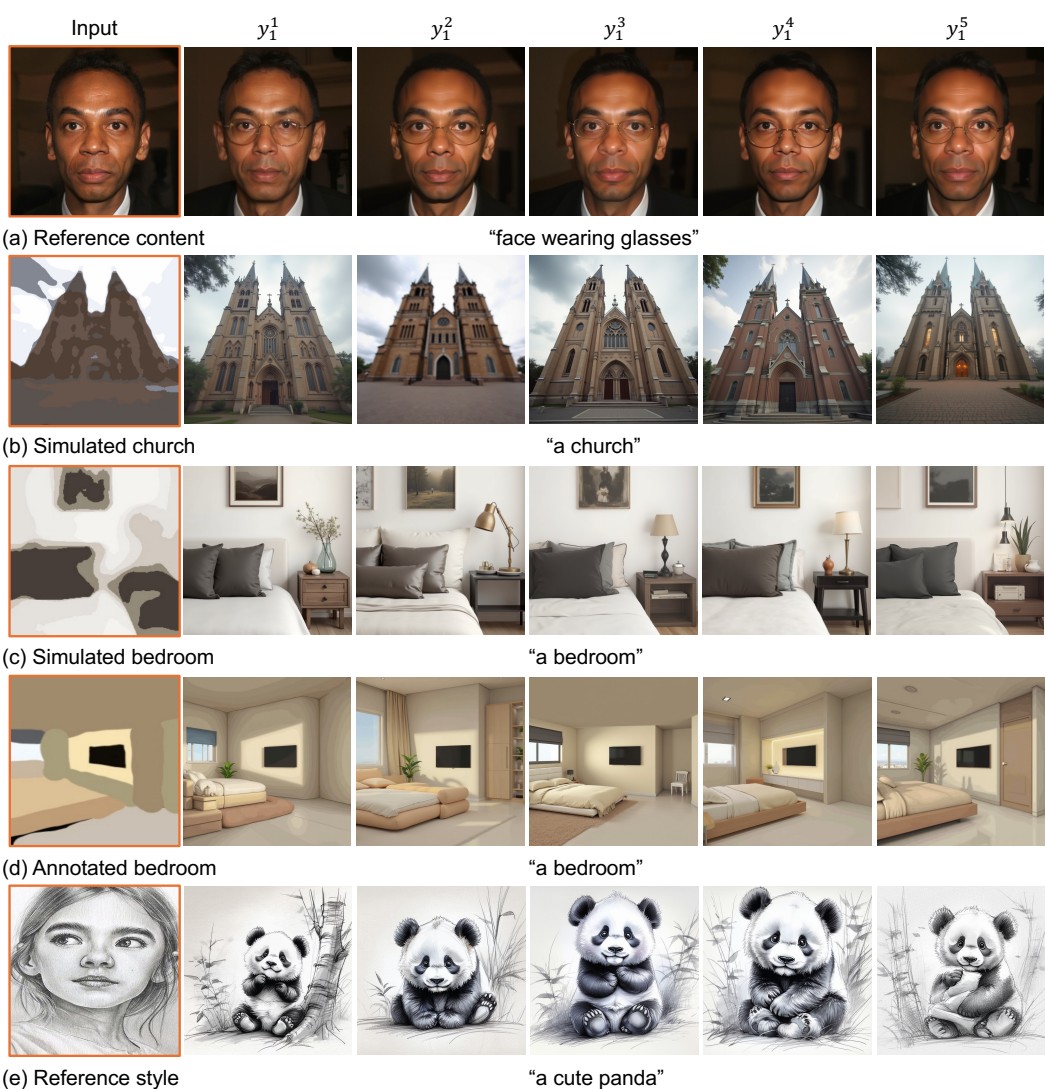

Figure 25: **Robustness to initialization ($\mathbf{y}_1$).** Given an input image (e.g., a style image of "pencil sketch" in (e)) as $\mathbf{y}_0$ and 5 different typical samples $\{\mathbf{y}_1^i\}_{i=1}^5$ from $p_1$, our method effectively captures the semantics of the reference input while adhering to the desired prompt (e.g., "a cute panda"). The annotated bedroom input in (d) is a screenshot of the bedroom from SDEdit (Meng et al., 2022). Compared to simulated stroke inputs in (b,c), the annotated stroke input in (d) helps with better alignment of the generated layout.

### D.15 COMPARISON WITH DDIM INVERSION FOR DIFFERENT START STEP

Figure 30 shows the effect of different starting time for DDIM Inversion. The generated samples become less realistic but more faithful as the starting time increases.

### D.16 LIMITATION

The lack of comparison with *expensive* diffusion-based editing solutions may be viewed as a limitation. However, these implementations are either not available for Flux or not directly applicable due to Flux's distinct multi-modal architecture. The key contribution of this paper lies in its theoretical foundations, validated using standard benchmarks and relevant baselines.

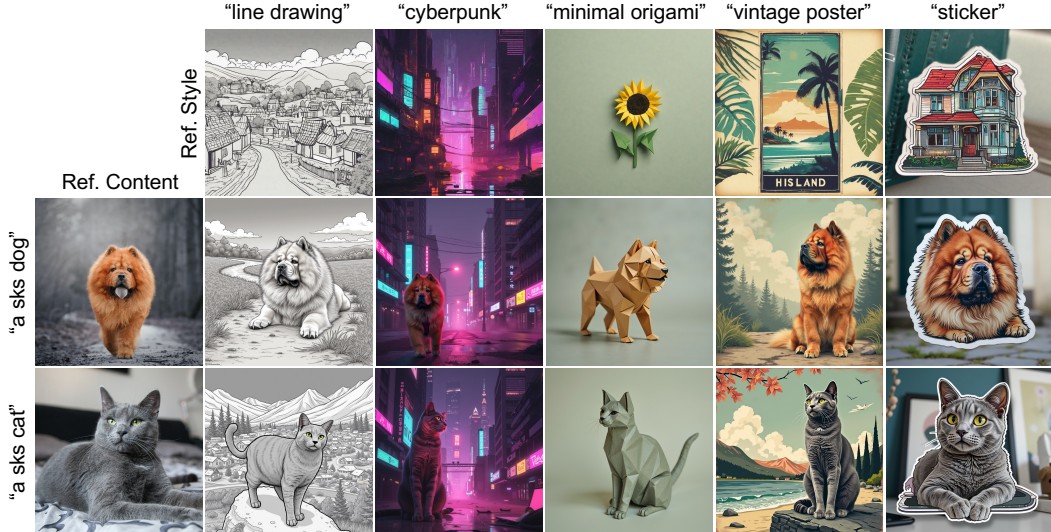

Figure 26: **Compatibility of our method with LoRA for content-style composition.** The generated image preserves the identity of the reference content while adhering to the desired style.

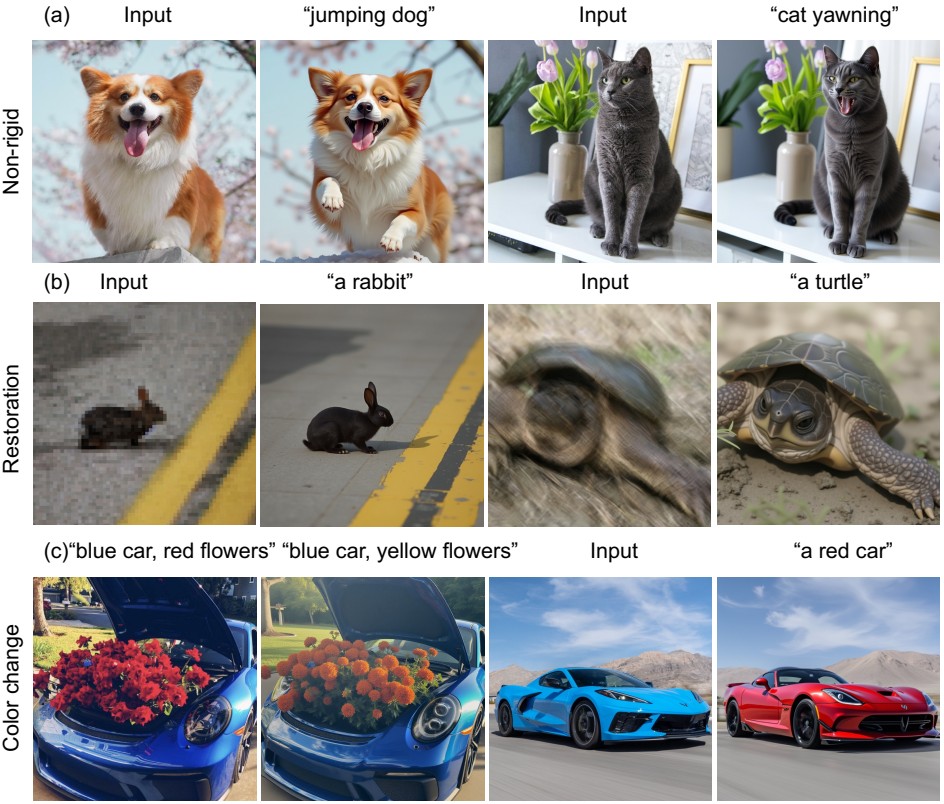

Figure 27: **Image editing in (a) non-rigid task, (b) image restoration and (c) local color change.** The proposed method generalizes to a wide-variety of inverse problems without training, test-time optimization or cross-attention manipulation.

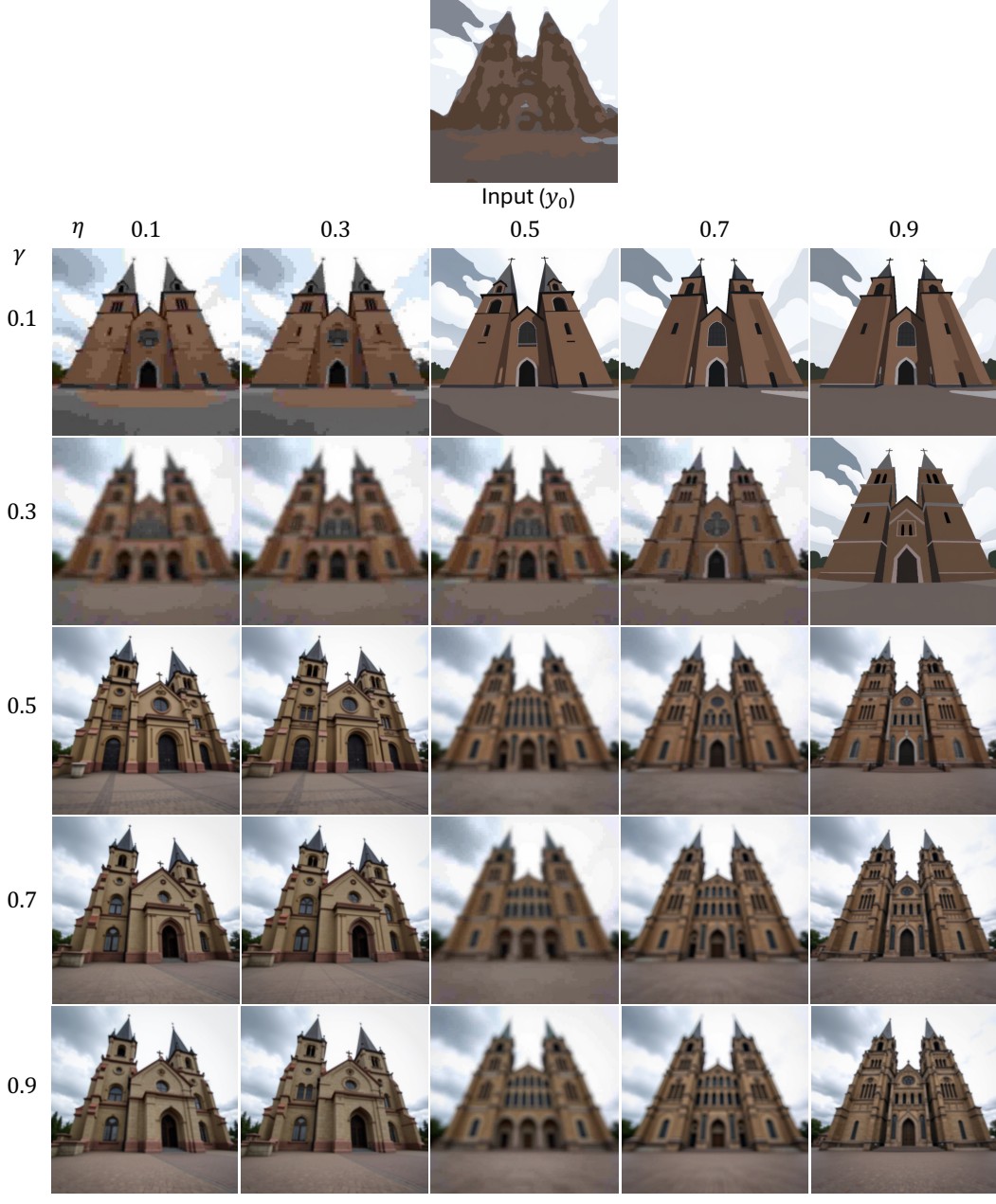

Figure 28: **Impact of the forward and reverse controller guidance in image editing.** Increasing $\gamma$ helps transform an "atypical" sample to a "typical" sample, making the generated samples more realistic. Subsequently, increasing the controller guidance for reverse flows ($\eta$) improves faithfulness to the reference input, as discussed in §3.

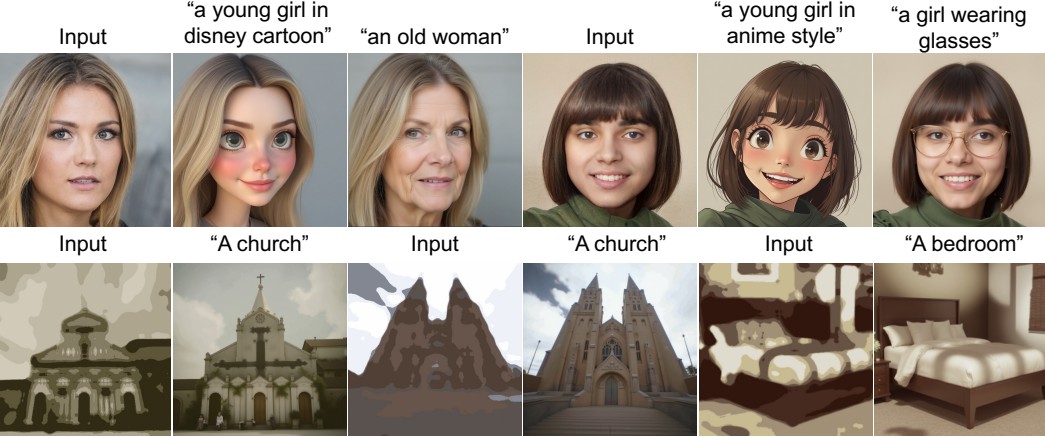

Figure 29: **Face editing and Stroke2Image generation in 8 steps.** We noise and denoise using a distilled model that is capable of sampling from the data distribution in 8 steps.

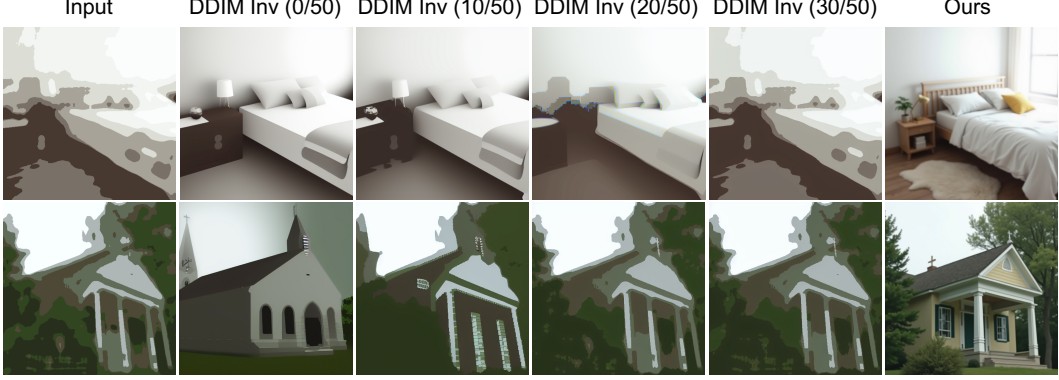

Figure 30: **Qualitative comparison with DDIM Inversion with different starting time.** Our method outperforms DDIM Inversion with a mid-way starting point.

