# OpenReview forum: "Semantic Image Inversion and Editing using Rectified Stochastic Differential Equations"
_ICLR.cc/2025/Conference — ICLR 2025 Poster_

### Official Review · Reviewer_QdiR · 2024-10-31

**Soundness:** 3
**Presentation:** 4
**Contribution:** 4
**Rating:** 8
**Confidence:** 4

**Summary:**

Given a generative ODE that links pairs of noise and image, one can solve the forward/reverse ODEs to invert an image to noise or reconstruct noise into an image.

This paper modify the drift term of ODE to guide the process to terminate at high-likelihood noise/image. The guidance term is derived from LQR problem, which of solution is a minimum energy path to convert "any" image (noise) to given sample of random noise (image). This guidance is equal to conditional vector field that is computed analytically, without any optimization.

The authors propose a controlled forward/reverse ODE with drift term computed by linearly interpolating the conditional vector field and the original vector field. By adjusting the interpolation coefficient, the model achieves a trade-off between high-likelihood solutions and consistency to the original image.

Furthermore, the paper interprets the controlled ODE within the context of stochastic differential equations (SDEs), highlighting the robustness of SDEs to initial conditions and offering a deeper insight into the proposed method.

Experimental results validate the method's effectiveness in inversion-editing tasks, showing improved performance over existing techniques.

**Strengths:**

- Theoretical analysis provides valuable insights for the proposed method.
- The proposed framework of controlled ODE provides a design space for other applications, by adding regulation to the conditioned vector field.
- Experimental analysis is well established and demonstrating the effectiveness of the proposed method compared to baselines.

**Weaknesses:**

- Missing comparison or discussion with fixed-point based inverse methods (ReNoise[1], AIDI[2]), which are applicable Euler solver. Their motivation is to reduce the error induced in inversion process with fixed point iterations and improve the editing performance.
- By design, controlled ODE will generate majority sample since it is guided toward mode. Thus, the proposed method may lose diversity even though we can control it via parameter $\gamma$. For example, edited faces always look at camera (Figure 5,6,7,10). But, this would be another open question.
- Proposed controlled forward ODE may break the OT path of Rectified Flow, as it compute conditioned vector field $u_t(Y_t|y_1)$ with fixed noise sample $y_1\sim \mathcal{N}(0, \mathrm{I})$.


**References**

[1] Effective real image editing with accelerated iterative diffusion inversion, ICCV 2023

[2] ReNoise: Real Image Inversion Through Iterative Noising, ECCV 2024

**Questions:**

- Have authors tried to restore blurry or downsampled image using the controlled forward/reverse ODE? Although it may not reconstruct the true signal exactly, it seems possible to get much clearer image.
- Have authors tried non-rigid editing task? (e.g. dog -> jumping dog)
- Have authors tried local editing (e.g. red flower and blue car -> yellow flower and blue car) rather than global context editing (e.g. man -> woman, photo -> pixar)?
- Have authors reduced the NFE lower than 28? The reviewer just wonder if fast sampling is available even with controlled vector field, which may break the OT path established by reflow. Just for the reference, [1] used fixed point iteration to make more accurate inversion process and demonstrated that the method enables real-time editing using Flux.1 with 4NFEs. This does not mean that comparison with [1] is required.

**Reference**

[1] Lightning-Fast Image Inversion and Editing for Text-to-Image Diffusion Models, arxiv 2023

---

> ### Author Response · Authors · 2024-11-21
> **Official Response by Authors**
>
> Dear Reviewer QdiR,
>
> Thank you for highlighting our **theoretical insights**, the **versatility** of the controlled ODE framework, and the **robustness** of our experimental analysis. We have revised the draft with new experimental details (colored green) to incorporate your suggestions. Below, we address your remaining questions.
>
> **(Q1) Missing comparison or discussion with fixed-point based inverse methods (ReNoise[1], AIDI[2]), which are applicable Euler solver. Their motivation is to reduce the error induced in inversion process with fixed point iterations and improve the editing performance.**
>
> (A1) Thank you for sharing ReNoise[1] and AIDI [2] with us. The accelerated iteration procedure from AIDI [2] and the renoising iteration step in ReNoise [1] are indeed valuable contributions that could enhance the inversion and editing capabilities of our method. With our theoretical and experimental results establishing a solid connection between diffusion models and rectified flows, we hope that incorporating creative sampling techniques developed for DDIM inversion, like ReNoise [1] and AIDI [2], would be an excellent direction for future research. We have added this discussion in the revised paper (Appendix C.8).
>
>
> **(Q2) By design, controlled ODE will generate majority sample since it is guided toward mode. Thus, the proposed method may lose diversity even though we can control it via parameter γ. For example, edited faces always look at camera (Figure 5,6,7,10). But, this would be another open question.**
>
> (A2) This is indeed an interesting open problem worth exploring in the future. We hope that our theoretical insights would be helpful in studying this problem as the community progresses towards rectified flows.
>
> **(Q3) Proposed controlled forward ODE may break the OT path of Rectified Flow, as it compute conditioned vector field ut(Yt|y1) with fixed noise sample y1∼N(0,I).**
>
> (A3) This is a reasonable proposition. However, breaking this OT path is essential to transforming “atypical’’ samples from $p_0$ into “typical samples” from $p_1$. Following the OT path would instead transform “atypical” samples from $p_0$ into “atypical” samples from $p_1$, which is not the desired outcome.
>
>
> **(Q4) Have authors tried to restore blurry or downsampled image using the controlled forward/reverse ODE? Although it may not reconstruct the true signal exactly, it seems possible to get much clearer image.**
>
> (A4) Thank you for the question. We have now added Figure 27 (b) to show restoration of a downsampled and blurry image using our controlled forward and reverse ODEs.
>
> **(Q5) Have authors tried non-rigid editing task? (e.g. dog -> jumping dog)**
>
> (A5) We have now included results for non-rigid editing tasks in Figure 27 (a).
>
> **(Q6) Have authors tried local editing (e.g. red flower and blue car -> yellow flower and blue car) rather than global context editing (e.g. man -> woman, photo -> pixar)?**
>
> (A6) We have now included results for local color change in Figure 27 (c). Figure 26 shows the compatibility of our method with popular parameter-efficient finetuning methods (e.g. LoRA), which can improve local editing in a desired scenario. We believe that the suggested local editing experiment is more suitable for mask-based editing algorithms, which is an interesting avenue for future research. To this end, we could invert only part of the image instead of inverting the whole image and apply masked controller guidance to change only the desired part of the image. Thanks for the question.
>
>
>
> **(Q7) Have authors reduced the NFE lower than 28? The reviewer just wonder if fast sampling is available even with controlled vector field, which may break the OT path established by reflow. Just for the reference, [3] used fixed point iteration to make more accurate inversion process and demonstrated that the method enables real-time editing using Flux.1 with 4NFEs. This does not mean that comparison with [3] is required.**
>
> (A7) We have not yet explored accelerating the sampling process with controlled vector fields, which seems a highly promising direction for future research. Inspired by your question, we have now included results for Turbo-Base LoRA in Figure 29 (Appendix C.13) to demonstrate semantic image editing in as few as 8 steps. It is remarkable that [3] could achieve real-time editing using Flux.1 with 4 NFEs.
>
> ### Concluding Remark
>
> We hope that the clarifications and additions above adequately address your concerns. We are committed to address any additional points you may have during the discussion phase.
>
> ### Reference
>
> [1] Effective real image editing with accelerated iterative diffusion inversion, ICCV 2023
>
> [2] ReNoise: Real Image Inversion Through Iterative Noising, ECCV 2024
>
> [3] Lightning-Fast Image Inversion and Editing for Text-to-Image Diffusion Models, arxiv 2023

---

> > ### Comment · Reviewer_QdiR · 2024-11-23
> > **Thanks for the response.**
> >
> > I appreciate the authors' efforts in providing additional results and adding discussions on other inversion methods. I will maintain my initial score.

---

> > > ### Author Response · Authors · 2024-11-23
> > > **Official Comment by Authors**
> > >
> > > Thank you for your prompt response. We are glad to hear that the additional experiments and the discussion on inversion methods in the revised draft addressed all your questions.

---

### Official Review · Reviewer_g1Xz · 2024-11-03

**Soundness:** 3
**Presentation:** 3
**Contribution:** 3
**Rating:** 5
**Confidence:** 4

**Summary:**

The paper proposes an "inversion" method for flow-based models, like Flux. The method allows to edit either natural images (from the distribution on which the model was trained) or unnatural images (like paint strokes). The inversion process balances between a forward process that inverts the reverse flow ODE (like in DDIM inversion) and a process that drives the inversion towards some fixed, random noise vector, so that the inversion lands in a region of typical realizations of the prior Gaussian distribution. Editing is done by starting from the noise obtained during the inversion and performing a reverse process that balances between the standard vector field and a vector field that pushes towards the input image, all while injecting the target text prompt.
Experiments are presented on several datasets and editing tasks, including quantitative evaluations and a user study, comparing to several baseline methods.
An additional contribution of the paper is the introduction of an SDE variant for the proposed Controlled Rectified Flow process, which as a special case provides an SDE variant for the standard flow sampling process. This allows sampling using a stochastic process, which is less sensitive to the initial noise, rather than a deterministic process, which is a deterministic function of the initial noise.

**Strengths:**

- The proposed Controlled Rectified Flow formulation is mathematical elegant.
- The SDE variant of the method (and of regular flow as a special case) is quite nice, though it is not explored experimentally.
- The paper is well written and easy to follow.

**Weaknesses:**

- The vast majority of the comparisons presented in the paper are against baselines implemented for different models (some of which are not even specified). Comparing between an editing method working with Flux and an editing method working with SD1.5, for instance, is meaningless. It does not reveal whether the differences in performance are due to the editing method or due to the fact that the underlying model is different (e.g. Flux is much stronger than SD1.5). The only exception is Fig. 7, which shows only two qualitative comparisons on Flux, but the baselines are very weak: SDEdit is known not to be consistent with the original image, and DDIM inversion is known to perform much better for editing when running the process only up to some intermediate t<1 (see e.g. Fig. 10 in [1]). I would expect to see at least the baseline of DDIM inversion for Flux going up to some midway point.
- The qualitative results are underwhelming, especially on LSUN Bedroom. Fig. 4 in the main text seems cherry picked, as in Figs. 13,15 in the appendix, none of the edited results look remotely similar to the input stroke image. It is not clear why this is not captured by the L2 metric in Table 1. The age and gender editing results are also quite weak. For instance, in Figs. 16,17, all edits look very far from the source.
- The proposed inversion process pushes towards some noise y_1, which is arbitrarily fixed a-priori. This means that the parameter eta interpolates between some fixed generic image corresponding to this y_1, and the original image, as seen in Fig. 3. However, the effect that the choice of this y_1 has on the editing is not explored. It seems to me that when the noise vector y_1 encodes a completely different geometric structure than the image being edited, the edited result may significantly deviate from the original image. On the other hands, when y_1 happens to encode a similar structure, the edited result may be closer to the original image. The paper doesn't provide an ablation study illustrating the effect of y_1 and doesn't provide a clever method to choose it.

[1] Huberman-Spiegelglas et al., "An edit friendly DDPM noise space: Inversion and manipulations", CVPR, 2024.

**Questions:**

I would be happy the hear the authors' thoughts on the weaknesses mention above.
In particular:
- What is the effect of y_1? How different are are results obtained with different y_1's? Is there a clever way to choose y_1?
- How does the method compare to DDIM inversion on Flux performed up to some midway timestep t? Namely, the inversion is up to some t*, and then the sampling starts from this t* using the text prompt describing the desired edit.
- How does the method work on other flow-based models, like SD3?
- Why does the L2 metric in Table 1 not capture the apparent discrepancies between the edited and original images in Figs. 13,15? As mentioned above, the edited images look almost completely unrelated to the original images.

---

> ### Author Response · Authors · 2024-11-21
> **Official Response 1/2 by Authors**
>
> Dear Reviewer g1Xz,
>
> Thank you for highlighting our work as **mathematically elegant**, the **SDE variant as quite nice**, and the paper as **well written and easy to follow**. We have revised the draft with new experimental details (colored green) to incorporate your suggestions. Below, we address your remaining concerns.
>
>
> **(Q1) The SDE variant of the method (and of regular flow as a special case) is quite nice, though it is not explored experimentally.**
>
> (A1) Thank you for your comment. We have now included Figures 22 and 23 in Appendix C.7 to demonstrate sampling via the SDE variant of regular flows.
>
>
> **(Q2) The vast majority of the comparisons presented in the paper are against baselines implemented for different models (some of which are not even specified). Comparing between an editing method working with Flux and an editing method working with SD1.5, for instance, is meaningless. It does not reveal whether the differences in performance are due to the editing method or due to the fact that the underlying model is different (e.g. Flux is much stronger than SD1.5). The only exception is Fig. 7, which shows only two qualitative comparisons on Flux, but the baselines are very weak: SDEdit is known not to be consistent with the original image, and DDIM inversion is known to perform much better for editing when running the process only up to some intermediate t<1 (see e.g. Fig. 10 in [1]). I would expect to see at least the baseline of DDIM inversion for Flux going up to some midway point.**
>
> **How does the method compare to DDIM inversion on Flux performed up to some midway timestep t? Namely, the inversion is up to some t, and then the sampling starts from this t using the text prompt describing the desired edit.**
>
> (A2) **[SD1.5 vs Flux]** SD1.5 is the base model used in their source code. We have reimplemented these methods using the same model (e.g. Flux) as ours. Therefore, the results with the same FLUX model in Figure 7 and Table 1-2 show that the improved performance is due to our method and not from the underlying model. Besides, we introduce a new fundamental method for rectified flows grounded in solid theoretical foundations, and hence it is reasonable to compare with other fundamental methods for diffusion models (e.g., SDEdit and DDIM inversion). As noted in Section 2 (L110-118) and in our limitations (L537-539), most of the existing methods are not directly applicable due to the MM-DiT architecture of Flux.
>
> **[Midway Point]**
> We thank the reviewer for pointing us to the very interesting result in Figure 10 of [1]. We added Figure 30 for experimenting with different midway starting points with DDIM inversion, showing that midway points fail to improve the results in the stroke2image task. Also, we compare DDIM inversion (Flux Inversion in Figure 7) with a midway starting point which is the same as ours ($t=1-6/28=0.78$ for glasses and $t=1-3/28=0.89$ for stroke2image). As discussed in Section 3.2 (L185-208), deterministic methods (e.g., DDIM Inversion, NTI, NTI+P2P, and Flux) transform “atypical” (corrupt) samples into “atypical” samples due to the lack of randomness in the noising process. This shows the benefit of our controllers ($\gamma$ and $\eta$), which are the differentiating factors between DDIM inversion and ours. We would be happy to discuss further if the reviewer has any more concerns.

---

> ### Author Response · Authors · 2024-11-21
> **Official Response 2/2 by Authors**
>
> **(Q3) The qualitative results are underwhelming, especially on LSUN Bedroom. Fig. 4 in the main text seems cherry picked, as in Figs. 13,15 in the appendix, none of the edited results look remotely similar to the input stroke image. It is not clear why this is not captured by the L2 metric in Table 1. The age and gender editing results are also quite weak. For instance, in Figs. 16,17, all edits look very far from the source.**
>
> **Why does the L2 metric in Table 1 not capture the apparent discrepancies between the edited and original images in Figs. 13,15? As mentioned above, the edited images look almost completely unrelated to the original images.**
>
> (A3) The L2 loss calculates pixel-wise difference that captures image structures and color patterns that could map to various concepts. For example, the white stroke areas could correspond to bedsheets, carpet, pillows or lighting through the windows. Besides, the color palette of the stroke image, which creates an illusion of depth, could influence human perception and create biases toward specific configurations. Take Figure 14 (LSUN-bedroom) as an example, we calculate the L2 loss of the 4 images for ours (last column) and SDEdit (second column) compared with the input stroke images (first column). **The L2 loss for ours are 82.71, 66.46, 88.11, 63.80, averaging 75.27. The results for SDEdit are 97.74, 87.82, 90.54, 50.29, averaging 81.60.** This demonstrates that our results achieve better faithfulness to the stroke image.
>
> Specifically, our results have similar image structures (e.g., window frames in image 2 and contours of the white areas in image 3) and color patterns (e.g., wall/floor colors). Note that there will be loss of the faithfulness in certain areas due to the tradeoff to realism.
>
> We agree with the reviewer that evaluating such tasks like stroke2image and face editing are challenging due to the lack of ground truth and subjectiveness of human perception. We conduct large-scale human evaluation in Table 1 to confirm the effectiveness of the proposed RF Inversion.
>
> **(Q4) The proposed inversion process pushes towards some noise y_1, which is arbitrarily fixed a-priori. This means that the parameter eta interpolates between some fixed generic image corresponding to this y_1, and the original image, as seen in Fig. 3. However, the effect that the choice of this y_1 has on the editing is not explored. It seems to me that when the noise vector y_1 encodes a completely different geometric structure than the image being edited, the edited result may significantly deviate from the original image. On the other hands, when y_1 happens to encode a similar structure, the edited result may be closer to the original image. The paper doesn't provide an ablation study illustrating the effect of y_1 and doesn't provide a clever method to choose it.**
>
> **In particular, What is the effect of y_1? How different are are results obtained with different y_1's? Is there a clever way to choose y_1?**
>
> (A4) The proposed inversion process not only pushes towards some noise $y_1$ but also constructs a nearly reversible path from $y_0$ to $y_1$ using $u_t(y_t)$ in ODE (8). Our method is robust to the initial choice of $y_1 \sim p_1$, as long as ODE (8) is followed to obtain the structured noise $Y_1$, as shown in the newly added Figure 25.
>
> Alternatively, one could optimize $y_1$​ to minimize a terminal cost, but this approach poorly preserves structure because the uncontrolled reverse process drifts away from the optimized trajectory. This is precisely the reason why continuous adjustments are necessary to steer the reverse process. While optimizing $y_t$ for all $t$ could address this issue, the associated computational cost increases significantly. In contrast, our method derives a closed-form controller to adjust the trajectory efficiently, starting from a typical $Y_1$ obtained from ODE (8).
>
>
> **(Q5) How does the method work on other flow-based models, like SD3?**
> (A5) Our method is compatible with other flow-based models, such as SD3 (v3.5), as demonstrated in the newly added Figure 24.
>
> ### Concluding Remark
>
> We hope that the clarifications and additions above adequately address your concerns. If you are satisfied, we kindly request you to consider revising your score. We are committed to address any additional points you may have during the discussion phase.
>
>
> ### Reference
>
> [1] Huberman-Spiegelglas et al., "An edit friendly DDPM noise space: Inversion and manipulations", CVPR, 2024.
>
> Thank you for pointing out the related work [1], which we have now cited in the revised draft.

---

> > ### Author Response · Authors · 2024-11-25
> > **A gentle reminder to respond to our rebuttal**
> >
> > Dear Reviewer g1Xz,
> >
> > We hope the above clarifications and the additional experiments in the revised draft sufficiently addressed your concerns. If you are satisfied, **we kindly request you to consider updating the score to reflect the newly added results and discussion**. We remain committed to addressing any remaining points you may have during the discussion phase.
> >
> > Best,
> >
> > Authors of Paper #1202

---

> ### Comment · Reviewer_g1Xz · 2024-11-25
>
> I thank the authors for the answers and the newly added figures.
>
> Regarding Flux inversion (DDIM inversion), you state in your answer that you used it with a midway starting point which is the same as yours. Are those the results shown in Fig. 7? (this is not mentioned in the text referring to Fig. 7).
>
> In any case, my overall impression remains that the proposed editing framework does poorly in terms of fidelity to the source image. I am also not convinced that the choice of $y_1$ is not important in tasks requiring structure preservation (the newly added Figure 25 illustrates the effect of $y_1$ only in style transfer, which is a task that by definition doesn't require structure preservation).
>
> I should say that the newly added Figures 24,25 show nice results for style transfer. I don't believe I have encountered methods achieving similar results without some sort of fine-tuning or test-time optimization. I believe the paper would have greatly benefited from putting more emphasis on tasks of this nature, instead of on tasks like stroke-to-image, in which the results are significantly less impressive.
>
> Considering the paper's current state, I still lean towards keeping my score of 5.

---

> > ### Author Response · Authors · 2024-11-26
> > **Discussion with Reviewer g1Xz**
> >
> > Dear Reviewer g1Xz,
> >
> > We are glad that you found our newly added stylization results quite nice. Below, we address your remaining points.
> >
> > **(Q1) Regarding Flux inversion (DDIM inversion), you state in your answer that you used it with a midway starting point which is the same as yours. Are those the results shown in Fig. 7? (this is not mentioned in the text referring to Fig. 7).**
> >
> > (A1) Yes, the results in Figure 7 use the same starting point as our method, as discussed in our response **(A2) [Midway Point]**. This has now been clarified in the experimental details (L1032–1035 in Appendix C). Thanks for pointing this out.
> >
> >
> > **(Q2) In any case, my overall impression remains that the proposed editing framework does poorly in terms of fidelity to the source image. I am also not convinced that the choice of $y_1$ is not important in tasks requiring structure preservation (the newly added Figure 25 illustrates the effect of $y_1$ only in style transfer, which is a task that by definition doesn't require structure preservation).**
> >
> > (A2) We would like to clarify that any typical sample $y_1$ from $p_1$ suffices for our controller.
> > The reviewer has suggested studying the impact of $y_1$ in structure preservation. To demonstrate this, we have now added more results in Figure 25, showcasing the effect of $y_1$​ on structure preservation beyond style transfer. **The structure of the person, bedroom, and church is consistently preserved across randomly sampled $y_1$.** Notably, in the church example, the left and right towers are angled toward the center, and the peaks consistently appear in similar areas of the generated samples (same for the painting and pillows in the simulated bedroom example). The stroke paint leads to diversity in the generated samples for different initializations.
> >
> > Regarding stroke-to-image quality, **the manually annotated clean stroke images used in SDEdit are not publicly available**. Therefore, we simulate stroke images following Section D.2 (Figures 30 and 31) in SDEdit, as this strategy was also used in their large-scale evaluation. The simulated stroke images are noisy which increases variability in the layout of the generated images. We also added one example in Figure 25 (d) using an annotated bedroom stroke input (by taking a screenshot of stroke paint from Figure 1 of SDEdit), **showing that the layout of the generated images are better aligned with a cleaner stroke input.**
> >
> > In summary, while the choice of $y_1$​ does have an impact, our controller ensures that the semantics of the reference input are still preserved for different random choices of $y_1$.
> >
> >
> > **(Q3) I should say that the newly added Figures 24,25 show nice results for style transfer. I don't believe I have encountered methods achieving similar results without some sort of fine-tuning or test-time optimization. I believe the paper would have greatly benefited from putting more emphasis on tasks of this nature, instead of on tasks like stroke-to-image, in which the results are significantly less impressive.**
> >
> > (A3) We are pleased to hear that Figures 24 and 25 demonstrate nice results for style transfer and that the reviewer has not encountered methods achieving similar results without fine-tuning or test-time optimization. Thank you for the suggestion to emphasize style transfer. We would like to clarify that our work does not focus only on stroke-to-image generation; we have conducted extensive experiments on stylization as well, as shown in Figures 8, 20, 24, 25, 26, and 29.
> >
> > We followed the evaluation protocol used in the compared baseline, SDEdit, which emphasizes stroke-to-image generation. To ensure a fair comparison, we included experiments on this task while also providing extensive evaluations for stylization. Our qualitative and quantitative results demonstrate that the proposed method significantly outperforms its diffusion counterparts without requiring additional training, test-time optimization, or complex attention processors.
> >
> > ### Concluding Remarks
> >
> > We hope the above clarifications and the previous experiments have successfully addressed all the concerns. **If satisfied, we kindly request the reviewer to consider updating the score to reflect the newly added results and discussion**. We would be glad to address any additional points during the discussion phase.

---

> > > ### Comment · Reviewer_g1Xz · 2024-11-26
> > >
> > > I thank the authors for the additional results.
> > >
> > > The additional results in Fig. 25 confirm my suspicion that $y_1$ has a significant influence on the structure of the edited image, which can override the similarity to the input image.
> > >
> > > Regarding the stylization results, I'd like to clarify that I referred to those in Fig. 24 and Figs. 8(a) and 8(b), in which the style is taken from the reference image and the content is described by the text prompt. This is a different task than the stylization shown in the other figures, where the content is taken from the reference image and the style is described by the text prompt. I consider the latter category of results unsuccessful. For example, in Fig. 20 the stylized images don't resemble the original images.
> > >
> > > I will keep my score.

---

> ### Author Response · Authors · 2024-11-27
> **Discussion with Reviewer g1Xz**
>
> Dear Reviewer g1Xz,
>
> Thank you for your prompt response. Please find our response to your remaining points below.
>
> **(Q1) The additional results in Fig. 25 confirm my suspicion that $y_1$ has a significant influence on the structure of the edited image, which can override the similarity to the input image.**
>
> (A1)  We think that our difference in opinion stems from the interpretation of the word “structure”. We interpret the phrase “preserving structure” as maintaining the broad outline or layout of the shape (e.g., the shape of a church, the layout of a bedroom, or the outline of a face) from the reference image. However, the reviewer's interpretation seems to go beyond this, requiring finer details to be preserved as well. Based on our interpretation, while $y_1$​ does influence fine structural details as rightly pointed out by the reviewer, changing $y_1$ does not significantly influence coarse structural elements (e.g., the peaks appear at nearly the same position and the towers are angled toward the center in Figure 25 (b)). We would be happy to clarify this distinction and make it more explicit in the revised version.
>
> We believe another possible reason for the difference in opinion could stem from the confusion between $Y_1$​ and $y_1$​, which are two distinct quantities. Specifically, $Y_1$​ is obtained by simulating our controlled ODE (8) from $t=0$ to $t=1$, whereas $y_1$​ is a typical sample drawn from $p_1$​ and remains fixed throughout this simulation process. While we are unsure if this might be a point of confusion, we have added this paragraph to clarify the notations and help with the discussion.
>
> **(Q2) Regarding the stylization results, I'd like to clarify that I referred to those in Fig. 24 and Figs. 8(a) and 8(b), in which the style is taken from the reference image and the content is described by the text prompt. This is a different task than the stylization shown in the other figures, where the content is taken from the reference image and the style is described by the text prompt. I consider the latter category of results unsuccessful. For example, in Fig. 20 the stylized images don't resemble the original images.**
>
> (A2) We appreciate the reviewer’s clarification and the feedback on content-style composition where content and style are described by two different images. We would like to clarify that the purpose of Figure 20 is not to demonstrate content-style composition, but rather to showcase various expressions of stylization where the content is expected to change based on the given prompt. For example, in Figure 20, the reference content is an image of **a woman**, and the desired prompt specifies **a young girl** in disney 3d cartoon style. Again, the goal of this experiment is not content preservation, but to illustrate cartoonization with various expressions.
>
> For content preservation, our experiment in Figure 26 demonstrates that the generated images closely resemble the original content (e.g., dog breed) while adopting the desired style (e.g., “line drawing”). In Figures 7, 8 (c), 14, 15, 17, 18, 20 and 25(a,b,c,d), the content is described by an image and the editing requirement (e.g., “wearing glasses”, “sleeping cat” or “a church”) is described by text prompt. We hope this distinction clarifies the intention behind these results and might address some of the reviewer’s concerns.
>
> ### Concluding Remarks
>
> If we understand the reviewer correctly, on one hand the reviewer appreciates several aspects of our work: “the controlled rectified flow formulation is mathematically elegant”, “the stochastic variant of rectified flows is quite nice”, “the paper is well-written and easy to follow”, and “the reviewer has not encountered methods achieving similar results without fine-tuning or test-time optimization.” On the other hand, the reviewer has concerns about structure preservation and content-style composition. Hopefully, our above discussion gives some additional context around these concerns.
>
> **We respect the reviewer’s perspective and sincerely appreciate the time dedicated to engaging with us during the discussion phase.**

---

### Official Review · Reviewer_Mj6G · 2024-11-03

**Soundness:** 3
**Presentation:** 3
**Contribution:** 2
**Rating:** 6
**Confidence:** 4

**Summary:**

This paper tackles the problem of image inversion and editing with Rectified Flow models such as Flux. In particular, the authors are interested in the case where the input image is corrupted, and where faithful inversion and reconstruction is not necessarily ideal. To this end, the authors introduce an optimal control-inspired guidance scale, which interpolates between the unconditional flow trajectory and one given by an optimal controller. This effectively controls the tradeoff between faithfulness and editability during inversion. Additionally, the paper derives the stochastic differential equation associated with the ODE of flow matching, providing additional insights. They demonstrate the effectiveness of their method on two tasks: stroke-to-image synthesis and image editing.

**Strengths:**

- The problem that the authors are solving is significant, as the literature is progressively moving towards RF models.
- The proposed solution is sound and the results show the effectiveness of the method. In particular, the key contribution of conditioning the reverse process on the input image $\mathbf{y}_0$ is new and interesting.
- The derivation of the SDE associated with the RF ODE (Eq. 10) provides theoretical  value regarding RF models.
- Overall, the writing quality of the paper is very good. The explanations are clear, the equations rigorous.

**Weaknesses:**

While the structure of the paper is clear, some parts seem to lack purpose. For example, the introduction of the LQR problem and the optimal controller in the method seems superficial, as the resulting path is simply a linear path between the input image $\mathbf{y}_0$ and a sampled noise $\mathbf{y}_1$ (which is similar to SDEdit). Similarly, while the derived SDE in Eq. 10 has theoretical value, it is not used in practice as suggested by the Algorithms 1 & 2 in appendix C. Finally, the joint effect of the controller guidance parameters $\gamma$ and $\eta$ is not clearly exposed in the paper (see questions).

**Questions:**

Regarding the inversion process (Section 3.2 & 3.3):

- I was confused by the implication of Proposition 3.1. Is it correct to understand that, out of the box, RF models are better than DM models for inversion? i.e., one can get almost perfect reconstruction in RF models by simply solving the ODE backward and then forward? This seems to be the case according to Table 5. If this is really the case, the problem to solve is actually more about editability than faithfulness. I would suggest the authors make it more explicit in the main paper by contrasting/comparing with DM models, where DDIM inversion is known to be sensitive to discretization errors.
- Proposition 3.2 states that the solution to the LQR problem is simply a linear path between the input image $\mathbf{y}_0$ and a randomly sampled (fixed) noise $\mathbf{y}_1$. This seems exactly like what SDEdit proposes. I was wondering if the optimal control perspective is different in any way? What additional insight or value does introducing the LQR problem bring compared to the straightforward linear interpolation explanation of SDEdit?
- The main result of the first part is shown in Eq. 8, where the authors introduce the *controller guidance* $\gamma$. Conceptually, is Eq. 8 essentially a linear interpolation between SDEdit (linear path solution to the LQR problem) and DDIM inversion (following the ODE backwards)? If so, wouldn’t it be more efficient to solve the ODE backward (normal inversion) and interpolate the result with a Gaussian noise at time $t=1$?

Regarding the SDE formulation (Section 3.4):

- Eq. 10 is the SDE formulation of Eq. 8. However, the stochastic part only relates to the unconditional ODE (from RF model), not the linear path conditioned on $\mathbf{y}_1$. Have the authors considered also making the linear path ODE stochastic? Intuitively, I assume that it would remove the need to fix the noise sample, as the SDE would converge to the entire Gaussian distribution.
- Around l.235, the authors mention that SDEs are more robust to initial conditions, which motivates the SDE formulation in Section 3.4. However, looking at Algorithms 1 & 2 in the appendix C, it seems that this SDE formulation is never used in practice (they only appear in C.3 for the numerical simulations). Why?

Regarding the Algorithm & Results (Section 4 & 5):

- What values of $\gamma$ and $\eta$ are used in Figure 2.?
- In Fig. 5, what models are used for SDEdit, DDIM inversion, NTI?
- The authors point out in the appendix that they use a fix value $\gamma=0.5$ through out the experiments. I wonder how does $\gamma$ affect the final result for a given $\eta$? It would be also interesting to have a two-axis grid comparison with varying $\gamma$ and $\eta$ to better understand the interplay between these two guidance parameters.

Miscellaneous:

- The caption in Fig. 10 says “The number below each figure denotes the starting time scaled by 28”, which is not true, since the number is the controller guidance value in this Figure.
- Typo at l.192: “we need to process**s**”.

---

> ### Author Response · Authors · 2024-11-21
> **Official Response 1/2 by Authors**
>
> Dear Reviewer Mj6G,
>
> Thank you for highlighting the **significance** of the problem, as the field is increasingly adopting RF models. We appreciate your recognition of the **soundness** of our solution, the **effectiveness** of our results, and the conditioning of the reverse process on the input image $y_0$​ as **new and interesting**. We are also glad you found the derivation of the SDE associated with the ODE (10) to provide **theoretical value**. Thank you for appreciating the **clarity**, **rigor**, and overall **writing quality** of the paper. We have revised the draft with new experimental details (colored green) to incorporate your suggestions. Below, we address your remaining questions.
>
>
> Regarding the inversion process (Section 3.2 & 3.3):
>
> **(Q1) I was confused by the implication of Proposition 3.1. Is it correct to understand that, out of the box, RF models are better than DM models for inversion? i.e., one can get almost perfect reconstruction in RF models by simply solving the ODE backward and then forward? This seems to be the case according to Table 5. If this is really the case, the problem to solve is actually more about editability than faithfulness. I would suggest the authors make it more explicit in the main paper by contrasting/comparing with DM models, where DDIM inversion is known to be sensitive to discretization errors.**
>
> (A1) Yes, the main message is that, out of the box, **RF models are better than DM for inversion.** Due to space limitation, we had deferred the discussion on that to Appendix C.3. We are committed to make it explicit in the main paper as suggested.
>
> **(Q2) Proposition 3.2 states that the solution to the LQR problem is simply a linear path between the input image y0 and a randomly sampled (fixed) noise y1. This seems exactly like what SDEdit proposes. I was wondering if the optimal control perspective is different in any way? What additional insight or value does introducing the LQR problem bring compared to the straightforward linear interpolation explanation of SDEdit?
>
> (A2) Thank you for the insightful question. The optimal controller offers two unique perspectives: (1) Given $t\in (0,1)$, SDEdit computes $y_t=y_0 \times e^{-t} + y_1 \times \sqrt{1-e^{-2t}}$, while our method computes $y_t = t \times y_1 + (1-t) \times y_0$, which is a minimum energy path from $y_0$ to $y_1$. Our method also has another vector field $u_t(y_t)$ updating $y_t$ to a new state, say $\hat{y}_t$, essential for recovering the original image $y_0$. The optimal controller solves the LQR problem to steer $\hat{y}_t$ towards $y_1$. (2) The optimal controller allows for a nearly reversible piecewise linear path from $y_0$ to $y_1$ that can closely track a non-linear trajectory in the limit. In contrast, SDEdit follows a strictly linear path from $y_0$ to $y_1$, which causes reconstruction errors for non-linear reverse paths as traced by DDPM inversion.
>
>
> **(Q3) The main result of the first part is shown in Eq. 8, where the authors introduce the controller guidance γ. Conceptually, is Eq. 8 essentially a linear interpolation between SDEdit (linear path solution to the LQR problem) and DDIM inversion (following the ODE backwards)? If so, wouldn’t it be more efficient to solve the ODE backward (normal inversion) and interpolate the result with a Gaussian noise at time t=1?**
>
> (A3) In our response (A2) we discussed the differences between SDEdit and our method. The DDIM inversion (following the ODE backwards) is equivalent to our unconditional vector field $u_t(y_t)$ for RFs. But these are two different ODEs with different drifts as given in Eq. 9 (set $\gamma=0$) and Eq. 14. Besides, the practical implications are significant because in DDIM inversion, only part of the drift (the score function) is approximated by a neural network, whereas in flows, the entire drift field is learned. Although theoretically equivalent, learning the full vector field in practice proves highly effective.
>
> Normal inversion combined with interpolating the result with Gaussian noise at $t=1$ is more efficient, as it avoids computing $u_t(y_t|y_1)$ for every $t\in[0,1]$. However, this results in poor reconstruction quality, as shown in RF Inversion ($\gamma = 0, \eta = 0.5$) (8->15) in Table 5. This corresponds to cases where $y_0$ is a typical sample from $p_0$, making interpolation with Gaussian noise at $t=1$ unnecessary. For atypical $y_0$​, however, interpolating only at the terminal time requires drastic changes to the terminal state. Gradual interpolation across all $t$ facilitates a smoother transition, which is also beneficial for the generation process.

---

> ### Author Response · Authors · 2024-11-21
> **Official Response 2/2 by Authors**
>
> Regarding the SDE formulation (Section 3.4):
>
> **(Q4) Eq. 10 is the SDE formulation of Eq. 8. However, the stochastic part only relates to the unconditional ODE (from RF model), not the linear path conditioned on y1. Have the authors considered also making the linear path ODE stochastic? Intuitively, I assume that it would remove the need to fix the noise sample, as the SDE would converge to the entire Gaussian distribution.**
>
> (A4) Thank you for the very elegant question. At present, we observe a potential issue with this approach: the linear path ODE is non-causal. To address this, it may be necessary to replace $y_1$ with the conditional expectation $E[Y_1|y_t]$, which can be computed using Tweedie’s formula via score function.
>
> We agree this is a compelling direction for future research, and we would greatly appreciate any thoughts the reviewer could share on constructing any alternate stochastic version of the linear path ODE. This could indeed provide a pathway to removing the need for a fixed noise sample while ensuring convergence to the entire Gaussian distribution.
>
>
> **(Q5) Around l.235, the authors mention that SDEs are more robust to initial conditions, which motivates the SDE formulation in Section 3.4. However, looking at Algorithms 1 & 2 in the appendix C, it seems that this SDE formulation is never used in practice (they only appear in C.3 for the numerical simulations). Why?**
>
> (A5) As noted around L235, the robustness of SDEs to initial conditions was studied in prior works. The SDE formulation in Section 3.4 offers insight into the trade-off between consistency with the corrupted image and the terminal invariant distribution (L241). Additionally, it enables the design of a stochastic sampler for Flux, as shown in the newly added Figures 22 and 23. This sampler is derived by reversing a special case of SDE (10) from Section 3.4.
>
> We use deterministic samplers for Flux in Algorithms 1 and 2, following the practice of preferring DDIM over DDPM inversion, to better preserve faithfulness. Studying stochastic samplers for Flux is an interesting avenue for future work.
>
>
> Regarding the Algorithm & Results (Section 4 & 5):
> **(Q6) What values of γ and η are used in Figure 2.?**
>
> (A6) In Figure 2, we used $\gamma=0.5$ and a linearly decreasing $\eta_t=\eta \times (1-t)$ where the strength $\eta=1.0$. For $\eta_t$, we use the start time $s=8$ and stop time $\tau=25$. We now included this information in Table 4.
>
> **(Q7) In Fig. 5, what models are used for SDEdit, DDIM inversion, NTI?**
>
> (A7) In Figure 5, we use SD1.5 for SDEdit, DDIM Inversion and NTI as provided in their source code. We also compare our method against SDEdit and DDIM Inversion with Flux backbone in Figure 7. We added new quantitative results in Table 1 for SDEdit with SDXL and SD3 as base models.
>
> **(Q8) The authors point out in the appendix that they use a fix value γ=0.5 throughout the experiments. I wonder how does γ affect the final result for a given η? It would be also interesting to have a two-axis grid comparison with varying γ and η to better understand the interplay between these two guidance parameters.**
>
> (A8) Figure 28 now includes a two-axis grid comparing the effects of varying $\gamma$ and $\eta$. This highlights the role of $\gamma$ in generating a “typical” structured noise and $\eta$ in preserving the structural details of the reference input, demonstrating the benefits of controller guidance.
>
>
>
> Miscellaneous:
>
> **(Q9) The caption in Fig. 10 says “The number below each figure denotes the starting time scaled by 28”, which is not true, since the number is the controller guidance value in this Figure.**
>
> (A9) Corrected this typo. Thanks for reading the paper carefully.
>
> **(Q10) Typo at l.192: “we need to processs”.**
>
> (A10) Fixed this typo in the revised draft.
>
> ### Concluding Remark
>
> We hope that the clarifications and additions above adequately address your concerns. If you are satisfied, we kindly request you to consider revising your score. We are committed to address any additional points you may have during the discussion phase.

---

> > ### Comment · Reviewer_Mj6G · 2024-11-25
> >
> > I thank the authors for addressing the majority of my concerns and updating the paper accordingly. However, a few things remain unclear to me.
> >
> > (A2) In the inversion process (from $y_0$ to $y_1$), I understand that the solution to the LQR problem is steering the original inverse ODE. However, my original question was more about the need to introduce the LQR problem in the first place, since the solution is a simple linear path between $y_0$ and $y_1$. As the authors point out, their method computes $y_t = t\times y_1 + (1-t) \times y_0$, which I agree is not strictly the same as SDEdit in the diffusion formulation. I would argue that it remains conceptually identical, and the authors themselves go on to say “SDEdit follows a strictly linear path from $y_0$ to $y_1$”. So my question would be more about **whether the LQR problem is strictly necessary to your method**? as from my understanding, the proposed method can be summarized by:
> >
> > - for inversion, we interpolate linearly between the inverse vector field of the RF model $u_t(y_t)$, and a linear path connecting $y_0$ to $y_1 \sim \mathcal N(0,I)$ through a parameter $\gamma$;
> > - and for reconstruction/editing, we interpolate linearly between the vector field of the RF model $v_t(x_t)$, and a linear path connecting $x_0$ to $x_1=y_0$ (the original image) through a time-varying parameter $\eta$.
> > Can the authors provide any additional insights on this?
> >
> >
> > (A3) First, I think the authors meant to point to the case RF Inversion $(\gamma=0.5, \eta=0)$ in Table 5 and not $(\gamma=0, \eta=0.5)$. Assuming this is the case, what I wanted to point out originally was that **for a constant $\gamma$** (which seems to be the case for all the applications shown here), **the solution to Eq. 8 seems to be a linear interpolation between the image $\mathbf{y}_0$ and the solution to the inverse ODE in Eq. 6**:
> >
> > $$Y_1  = Y_0 + \int_0^1 dY_t$$
> > $$= Y_0 + \int_0^1 \left[ (1-\gamma) u_t(Y_t)  + \gamma u_t(Y_t|\mathbf{y}_1) \right] dt$$
> > $$=  Y_0 +  (1-\gamma) \int_0^1 u_t(Y_t)  + \gamma \int_0^1 u_t(Y_t|\mathbf{y}_1) dt$$
> > $$=  (1-\gamma) \left[ Y_0 + \int_0^1 u_t(Y_t)\right]  + \gamma \left[ Y_0 + \int_0^1 u_t(Y_t|\mathbf{y}_1) dt \right] $$
> > $$=  (1-\gamma) [{\textrm{solution to Eq. 6}}]  + \gamma [\mathbf y_1]$$
> >
> > If this is correct, Eq. 8 seems overcomplicated, as a linear interpolation of the final result of ‘following the ODE backward’ with a random noise sample $\mathbf{y}_1$ would achieve the exact same result.
> >
> > I hope I was able to clarify my questions, and would appreciate if the authors can share their opinion on this, as I might have misunderstood some aspects of the paper. Overall, I thank the authors for the replies and the thorough rebuttal, and am satisfied with the answers provided to the other questions.

---

> > > ### Author Response · Authors · 2024-11-25
> > > **Discussion with Reviewer Mj6G**
> > >
> > > Dear Reviewer Mj6G,
> > >
> > > Thank you for your timely feedback on our rebuttal. We are pleased to hear that **our clarifications and the revised draft addressed the majority of your concerns**. Below, we address your remaining comments.
> > >
> > > **(A2)** The reviewer is absolutely correct that the proposed approach can be viewed as a linear interpolation between the corresponding vector field and the linear path, which suffices for practical purposes. However, the optimal controller perspective offers valuable theoretical insights. Specifically, the base generative model can be interpreted as solving an optimal control problem with KL divergence as the terminal cost. Analogously, we demonstrate that the guided flow corresponds to a conditional optimal control problem with a modified terminal cost. Our controlled rectified flow then interpolates between these two vector fields, each derived from a distinct optimal control problem. This perspective provides a more principled foundation for algorithm design, potentially guiding future research.
> > >
> > > While we acknowledge the similarities between SDEdit and our approach, there are **two key conceptual differences**. **First**, SDEdit relies on a one-step interpolation, making it unclear how it handles perturbations in the drift caused by another vector field. In contrast, our approach corrects such perturbations dynamically through a new straight-line interpolation from the perturbed point to the original destination. This action enhances the realism of generated samples, while the reversal of the base generative model maintains faithfulness. **Second**, the reverse process in SDEdit is the standard DDPM, whereas ours employs a controlled rectified flow, offering a similar dynamic correction interpretation as described above.
> > >
> > > **(A3)** RF-Inversion  ($\gamma=0.5, \eta=0$) and ($\gamma=0,\eta=0.5$) in Table 5 indicate the importance of $\gamma$ and $\eta$ to the corresponding forward and reverse flows, respectively. The newly added Figure 28 also highlights their importance in a practical setting, thanks for the suggestion.
> > >
> > > We appreciate the reviewer's effort in clarifying the question. The derivation appears correct. There are two flows involved in this process: (1) the forward flow, which maps an image to a noise sample, and (2) the reverse flow, which maps a noise sample back to an image. For the forward flow, Eq. (8) can indeed be simplified as the reviewer suggested. However, for the reverse flow, implementing this simplified approach poses a challenge. Simulating the reverse flow ODE (1) until the end of the process ($t=1$) and interpolating the terminal state with the reference image $y_0$​ often results in a superposition of conflicting images, leading to blurry reconstructions or inconsistent styles.
> > >
> > > In contrast, our method progressively interpolates between two vector fields, ensuring a smooth transition that produces high-quality image generation and consistent stylization. Thus, our controlled rectified flows lead to smooth curved paths. Since our primary goal is reverse flow, we describe the forward flow in the same unified framework (smooth interpolation between vector fields).
> > >
> > > ### Concluding Remarks
> > >
> > > We hope the above clarifications and the previous experiments have successfully addressed all your concerns. If you are satisfied, we kindly request you to consider updating your score to reflect the newly added results and discussion. We would be glad to address any additional points you may have during the discussion phase.

---

> > > > ### Author Response · Authors · 2024-12-02
> > > > **Discussion with Reviewer Mj6G**
> > > >
> > > > Dear Reviewer Mj6G,
> > > >
> > > > As the discussion phase is approaching its end, we kindly request the reviewer to let us know if the above clarifications and the previously added experiments have addressed the remaining questions. We would be happy to address any additional points the reviewer may have during the remaining time of the discussion phase.
> > > >
> > > > We thank the reviewer for engaging with us in the discussion.
> > > >
> > > > Best,
> > > >
> > > > Authors of Paper #1202

---

> > > > > ### Comment · Reviewer_Mj6G · 2024-12-02
> > > > > **Thank you for the clarification**
> > > > >
> > > > > Dear authors,
> > > > >
> > > > > Your answer to my previous questions did help clarify my remaining interrogations. I thank the authors for their patience and their time, and will keep my positive score regarding the paper.
> > > > >
> > > > > Best,

---

### Official Review · Reviewer_XYr8 · 2024-11-03

**Soundness:** 3
**Presentation:** 2
**Contribution:** 3
**Rating:** 8
**Confidence:** 5

**Summary:**

This paper propose RF inversion using dynamic optimal control derived via a linear quadratic regulator. We prove that the resulting vector field is equivalent to a rectified stochastic differential equation.

**Strengths:**

This paper propose RF inversion using dynamic optimal control derived via a linear quadratic regulator. We prove that the resulting vector field is equivalent to a rectified stochastic differential equation.

**Weaknesses:**

1. The paper lacks a clear pipeline diagram, algorithm, or pseudocode, making it challenging to understand the methodology. The reviewer suggests adding an algorithm or pipeline diagram to improve clarity and readability.

2. The paper does not include a comprehensive comparison on established benchmarks for text-guided image editing. The sole focus on face editing in SFHQ for the “wearing glasses” task is insufficient to demonstrate the method's effectiveness. The authors are encouraged to refer to recent surveys or the latest papers in the field and to identify one or two well-recognized benchmarks for text-controlled image editing for a more complete evaluation. It would be beneficial to include comparisons with state-of-the-art image editing methods from 2024, rather than relying on older methods.

3. The proposed method involves a large number of hyperparameters, which is a potential drawback. The authors should perform a complete sensitivity analysis on all hyperparameters, as most training-free methods are known to be highly sensitive to hyperparameter choices, with different settings leading to vastly different editing results. The reviewer suggests specifying the hyperparameter values used for each case to provide better transparency.

**Questions:**

1. *In Equation (8), since the value of \( y_1 \) is unknown, how is the conditional term in Equation (8) calculated when \( y_1 \) is needed?*

2. *Similarly, in Equation (15), the calculation relies on \( y_0 \). This implies that the target \( y_0 \) is fully known, which raises fairness concerns. Reconstructing a value that is already available seems unfair and could be viewed as “cheating,” making a 100% reconstruction accuracy almost inevitable. Many training-free methods do not require access to \( y_0 \) during sampling. The authors should explain why they have chosen to compare reconstructions in this potentially biased manner.*

---

> ### Author Response · Authors · 2024-11-21
> **Official Response by Authors**
>
> Dear Reviewer XYr8,
>
> Thank you for highlighting that this paper proposes RF inversion using dynamic optimal control and proves that the resulting vector field is equivalent to a stochastic differential equation. We have revised the draft with new experimental details (colored green) to incorporate your suggestions. Below, we address your remaining concerns.
>
> **(Q1) The paper lacks a clear pipeline diagram, algorithm, or pseudocode, making it challenging to understand the methodology. The reviewer suggests adding an algorithm or pipeline diagram to improve clarity and readability.**
>
> (A1) We will work on a new Figure 1 for the pipeline in the camera-ready version. We provided algorithm descriptions in Section 4, and the pseudocode in Algorithms 1 and 2 (L1080-1114 in Appendix C). For added clarity, we have now explicitly referenced these algorithms in the main paper (L349-351).
>
> **(Q2) The paper does not include a comprehensive comparison ... relying on older methods.**
>
> (A2) As noted in Section 2 (L111-119) and in our limitations (L534-537), current state-of-the-art methods are incompatible with Flux. Nonetheless, we provide comparisons with leading training-free and training-based methods in Figures 5 (main draft), 14 and 15 (Appendix C.4). Since we introduce a new fundamental method for rectified flows grounded in solid theoretical foundations, it is reasonable to compare with other fundamental methods for diffusion models (e.g., SDEdit and DDIM inversion). First, we compare our method with these methods using their default base model (which is a diffusion model)  and then reimplement these methods using the same base model as ours (which is a rectified flow model). These experiments highlight that the improved performance stems from our algorithmic contributions.
>
> We have now included more results demonstrating the overall capabilities of our method RF-Inversion in Figure 8, and additional editing results on more established benchmarks like DreamBooth dataset in Figure 26.
>
> **(Q3) The proposed method involves a large number of hyperparameters, which is a potential drawback. The authors should perform a complete sensitivity analysis on all hyperparameters, as most training-free methods are known to be highly sensitive to hyperparameter choices, with different settings leading to vastly different editing results. The reviewer suggests specifying the hyperparameter values used for each case to provide better transparency.**
>
> (A3) As detailed in Appendix C.2, our method introduces only 3 hyper-parameters: starting time ($s$), stopping time ($\tau$), and controller guidance strength ($\eta$). Appendices C.1, C.2, and C.3 include a complete sensitivity analysis of all hyperparameters. The hyperparameter values for each experiment have been specified in Table 4. Figure 28 now includes a two-axis grid comparing the effects of varying $\gamma$ and $\eta$.
>
> **(Q4) In Equation (8), since the value of ( y_1 ) is unknown, how is the conditional term in Equation (8) calculated when ( y_1 ) is needed?**
>
> (A4) Lowercase $y_1$ denotes a realization of the random variable $Y_1$, meaning it is known at $t=0$ (the beginning of the process). It is independent of the given image $y_0$, drawn from $p_1$ at $t=0$, and remains fixed throughout the process.
>
> **(Q5) Similarly, in Equation (15), the calculation relies on ( y_0 ). This implies that the target ( y_0 ) is fully known, which raises fairness concerns. Reconstructing a value that is already available seems unfair and could be viewed as “cheating,” making a 100% reconstruction accuracy almost inevitable. Many training-free methods do not require access to ( y_0 ) during sampling. The authors should explain why they have chosen to compare reconstructions in this potentially biased manner.**
>
> (A5) We would like to clarify that $y_0$ is the given reference image, **not the target image**, so there is no fairness concern. Many training-free methods (e.g., SDEdit, DDIM inversion, NTI, LEDIT++), in fact, require access to $y_0$ to guide the reverse process. We clarify this with an example below.
>
> In the newly added Figure 8, the reference content image of a cat in (c) is denoted by $y_0$. The target image—a tiger—is unknown, so achieving a 100% reconstruction of $y_0$ is not the desired goal. In the stroke-to-image generation task, NTI and NTI+P2P suffer because they explicitly optimize for $100\%$ reconstruction, as shown in Figure 14 and 15 in Appendix C.4. The  goal is to stay faithful to $y_0$ while applying desired edits (e.g.,  transforming a cat into a tiger or converting a stroke paint into a realistic image), as described in the introduction (L26-30).
>
> ### Concluding Remark
>
> We hope that the clarifications and additions above adequately address your concerns. If you are satisfied, we kindly request you to consider revising your score. We are committed to address any additional points you may have during the discussion phase.

---

> > ### Comment · Reviewer_XYr8 · 2024-11-26
> > **Reviewer's feedback**
> >
> > Thanks for the response that the authors made. Most of my concerns are addressed, hence I increased my rating. I wish the authors could release the code to contribute to the community, if possible.

---

> > > ### Author Response · Authors · 2024-11-26
> > > **Thank you for increasing your rating**
> > >
> > > Dear Reviewer XYr8,
> > >
> > > We thank you for increasing your rating and are pleased to hear that most of your concerns have been successfully addressed. We are happy to provide access to the code after the ICLR review cycle is complete.
> > >
> > > Best,
> > >
> > > Authors of Paper #1202

---

### Comment · Area_Chair_4izB · 2024-11-25
**Please check the authors' responses**

Dear reviewers,

Could you please check the authors' responses, and post your message for discussion or changed scores?

best,

AC

---

### Meta-Review · Area_Chair_4izB · 2024-12-22

**Metareview:**

This paper works on the image inversion and editing by using the stochastic equivalents of rectified flow models. The paper proposed the inversion of rectified flow method by dynamic optimal control via a linear quadratic regulator, and the resulting vector field is a rectified SDE. The proposed approach was applied to the zero-shot image inversion and editing, showing its effectiveness.  The strength of this work is on tackling the task of RF inversion using dynamic optimal control, resulting in rectified SDE. The reviewers raised some major concerns, e.g., the dynamic control by linear quadratic regulator is just a linear interpolation path resembling SDEdit (Reviewer Mj6G), the effect of  y_1 on the structure preservation of generation results (Reviewer g1Xz). These concerns are responded in the discussion phase, e.g., the rebuttal compares with the SDEdit, and the effect of y_1 is responded by adding new figures to show the structure preservation. The final decisions of the reviewers are scores of 8,6, 5, 8, and the reviewer g1Xz rated score of 5 under the acceptance bar, which seems that the reviewer was not fully satisfied with the responses. Overall, the paper received mostly positive final decisions after rebuttal. The paper can be accepted considering its strength in the RF inversion from dynamic optimal control perspective. However, the authors should carefully consider the remaining unsatisfactory points of reviewers, and also the relation to the previous related works in the resulting interpolation path and SDE dynamic in the final version.

**Additional Comments On Reviewer Discussion:**

Reviewer XYr8 raised questions on the lack of clear pipeline diagram, comparisons with established benchmarks for text-guided image editing, and involving large number of hyper-parameters.  Reviewer Mj6G commented on the concerns of resulting linear path resembling SDEdit, comparisons with SDEdit, and details on the SDE formulations, algorithm & results, etc. Reviewer g1Xz raised questions on the effect of y_1, comparison to DDIM inversion on Flux performed up to some midway timestep, work on other flow-based models, structure preservation in the edited image, etc. Reviewer QdiR suggested to compare with fixed-point based inverse methods (ReNoise[1], AIDI[2]), concerned that may lose diversity and break the OT path of Rectified Flow, etc. The responses mostly addressed the concerns of reviewers. The reviewers XYr8, Mj6G, QdiR are satisfied with these responses and rated positive final scores, and reviewer g1Xz kept the score of 5.

---

### Decision · Program_Chairs · 2025-01-22

Accept (Poster)